# MOBILE-GS: REAL-TIME GAUSSIAN SPLATTING FOR MOBILE DEVICES

**Xiaobiao Du** [1,3]   **Yida Wang** [3]   **Kun Zhan** [3]   **Xin Yu** [2]*

[1] University of Technology Sydney   [2] Adelaide University   [3] Li Auto Inc.

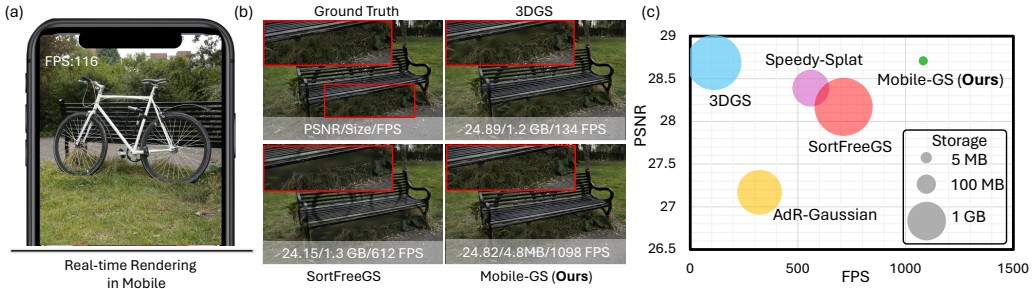

Figure 1: **Mobile-GS** is the first real-time Gaussian Splatting method that can reach **116** FPS rendering speed in the $1600 \times 1063$ resolution for Bicycle on the mobile equipped with the Snapdragon 8 Gen 3 GPU as shown in **(a)**. We evaluate rendering quality, storage costs, and inference speed on an RTX 3090 Ti GPU in **(b) and (c)**. Our Mobile-GS integrates depth-aware order-independent rendering, compression, and distillation techniques to deliver comparable rendering quality compared with the original 3DGS, while substantially reducing the storage requirements to **4.8 MB** and achieving **1098 FPS** on the unbounded scene, thereby enabling efficient deployment on mobile devices.

## ABSTRACT

3D Gaussian Splatting (3DGS) has emerged as a powerful representation for high-quality rendering across a wide range of applications. However, its high computational demands and large storage costs pose significant challenges for deployment on mobile devices. In this work, we propose a mobile-tailored real-time Gaussian Splatting method, dubbed Mobile-GS, enabling efficient inference of Gaussian Splatting on edge devices. Specifically, we first identify alpha blending as the primary computational bottleneck, since it relies on the time-consuming Gaussian depth sorting process. To solve this issue, we propose a depth-aware order-independent rendering scheme that eliminates the need for sorting, thereby substantially accelerating rendering. Although this order-independent rendering improves rendering speed, it may introduce transparency artifacts in regions with overlapping geometry due to the scarcity of rendering order. To address this problem, we propose a neural view-dependent enhancement strategy, enabling more accurate modeling of view-dependent effects conditioned on viewing direction, 3D Gaussian geometry, and appearance attributes. In this way, Mobile-GS can achieve both high-quality and real-time rendering. Furthermore, to facilitate deployment on memory-constrained mobile platforms, we also introduce first-order spherical harmonics distillation, a neural vector quantization technique, and a contribution-based pruning strategy to reduce the number of Gaussian primitives and compress the 3D Gaussian representation with the assistance of neural networks. Extensive experiments demonstrate that our proposed Mobile-GS achieves real-time rendering and compact model size while preserving high visual quality, making it well-suited for mobile applications. Project Page: https://xiaobiaodu.github.io/mobile-gs-project/

---

*Corresponding author: xin.yu@adelaide.edu.au

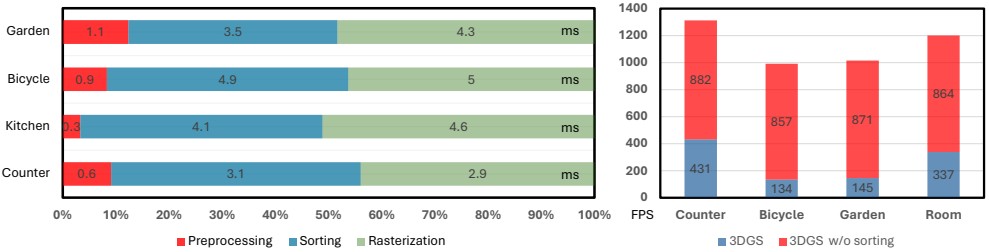

Figure 2: **Sorting as the primary performance bottleneck. Left:** Runtime analysis of the original 3DGS highlights that the sorting operation incurs a significant computational overhead during inference. **Right:** Removing the sorting step substantially accelerates 3DGS, achieving several-fold speedup compared to the original implementation.

## 1 INTRODUCTION

Neural Radiance Field (Mildenhall et al., 2021) is the first to leverage volume rendering for high-quality novel view synthesis (Barron et al., 2021; Wang et al., 2025), which has been applied to practical applications, like self-driving (Du et al., 2024a;b) and relighting (Wu et al., 2023). 3D Gaussian Splatting (3DGS) (Kerbl et al., 2023) is a recently introduced technique for high-quality 3D reconstruction that represents scenes as a set of anisotropic 3D Gaussian primitives. In contrast to traditional mesh- or voxel-based representations (Sitzmann et al., 2019; Shrestha et al., 2021; Tsalicoglou et al., 2023; Liu et al., 2020), Gaussian splatting leverages the continuous and differentiable nature of 3D Gaussians, enabling photorealistic rendering, precise novel view synthesis, and high-fidelity reconstruction. However, deploying Gaussian splatting on mobile platforms for real-time rendering remains challenging. The computational overhead of rendering tens of thousands of Gaussians, especially with the view-dependent effects, exceeds the capabilities of most modern mobile GPUs. This limitation highlights the pressing need for efficient solutions to enable real-time Gaussian splatting on resource-constrained platforms, such as smartphones and AR headsets.

Several lightweight 3D Gaussian Splatting methods, such as Scaffold-GS (Lu et al., 2024), Mini-Splatting (Fang & Wang, 2024), SplatFacto (Tancik et al., 2023), and C3DGS (Niedermayr et al., 2023) improve efficiency through pruning and compact representations. However, these methods rely on the traditional alpha blending, which requires a sorting process to render 3D Gaussians in the near-to-far order. We observed that this sorting process is the primary computational bottleneck, as shown in Fig 2, impeding real-time rendering on mobile devices. Therefore, to achieve real-time rendering performance on such platforms, there are several critical factors: (1) **Order-free rendering**: eliminating the time-consuming sorting process; (2) **Quantization**: compressing 3D Gaussians to reduce memory and bandwidth consumption; (3) **Fewer Gaussian points**: reducing the number of primitives while preserving visual quality.

In this work, we propose a real-time Gaussian Splatting method tailored for mobile devices, named Mobile-GS. As shown in Fig. 1, our proposed Mobile-GS can reach 116 FPS rendering speed on the mobile device with a Snapdragon 8 Gen 3 GPU, demonstrating real-time rendering speed on mobile devices. Our proposed Mobile-GS contains the following key components: (1) **Depth-aware Order-independent Rendering**: To circumvent the computationally intensive sorting process inherent in traditional alpha blending, we propose a depth-aware order-independent rendering technique, enabling faster rendering. To be specific, we discard the original alpha blending paradigm, which relies on the sorted 3D Gaussians. We propose a depth-aware weighting strategy that order-independently blends all related 3D Gaussians to the pixel. This strategy explicitly decreases the weight of the far 3D Gaussians and increases the significance of the near ones, enabling real-time performance, avoiding the sorting process. Although the proposed depth-aware order-independent rendering facilitates real-time rendering, the unordered blending can introduce transparency artifacts. To mitigate this, we further propose a neural view-dependent enhancement strategy that leverages a neural network conditioned on 3D Gaussian attributes and viewing direction to further capture view-dependent information. In this way, the rendering quality can be significantly improved, especially for view-dependent effects. (2) **First-order Spherical Harmonics Distillation**: Since the original 3DGS uses the third-order spherical harmonic (SH) function to represent appearance, it introduces numerous parameters and increases the storage burden. Therefore, we introduce a spherical

harmonic distillation technique to distill the first-order SH parameters under the guidance of the pre-trained teacher model, thus achieving faster rendering speed and lower storage usage. (3) **Neural Vector Quantization** : To deploy 3DGS on mobile devices, the quantization process is necessary to largely reduce storage usage and improve rendering speed. Herein, we introduce a neural vector quantization technique to quantize 3D Gaussian parameters grouped by K-means and compress the distilled SH features using lightweight neural decoders, thereby substantially minimizing overall storage costs. (4) **Contribution-based Pruning** : We also propose a contribution-based pruning strategy to prune redundant Gaussians according to their opacity and scale attributes. We reckon that the Gaussian with low opacity and scale indicates an insignificant contribution. With our pruning strategy, we can remove numerous Gaussians and further decrease storage costs.

Extensive experiments are conducted to qualitatively and quantitatively validate the effectiveness of our proposed Mobile-GS. We demonstrate that our method, deployed on mobile devices, achieves real-time rendering speed. Notably, our proposed Mobile-GS achieves high-quality and visually pleasing novel view synthesis, comparable to the original 3DGS, demonstrating that our approach can reliably reconstruct and render high-fidelity views. Our method consistently surpasses previous lightweight approaches, achieving state-of-the-art rendering speed and visually pleasing quality.

## 2 RELATED WORK

**3D Gaussian Splatting.** The recent 3D Gaussian Splatting (3DGS) technique (Kerbl et al., 2023), along with its numerous variants (Fang & Wang, 2024; Du et al., 2026; Lin et al., 2025; Bulo et al., 2025; Feng et al., 2025), employs anisotropic 3D Gaussians to represent scenes and leverages a tile-based differentiable rasterizer to render novel views. MVGS (Du et al., 2026) is the first to propose multi-view learning to enhance the multi-view constraint of 3DGS in the optimization stage, which significantly improves the holistic rendering performance. Scaffold-GS (Lu et al., 2024) introduces a hierarchical scaffold structure to reduce the number of Gaussians for high-quality rendering, while maintaining visual fidelity. Mini-Splatting (Fang & Wang, 2024) focuses on pruning and densification strategies to produce highly compact Gaussian structures. These methods facilitate high-resolution rendering while preserving high visual fidelity. Subsequently, more and more methods (Chen et al., 2024b; 2025; Wang et al., 2024b; Liu et al., 2024) are proposed to increase the compression ratio for more lightweight representations. However, these approaches require rendering Gaussians in a particular order, typically determined by depth through a sorting process. This depth-sorting process introduces multiple challenges, including increased implementation complexity and the potential for visual artifacts, such as abrupt texture variations and popping artifacts, as discussed in (Radl et al., 2024). In particular, we found that the computational overhead introduced by sorting is very serious, which significantly impedes real-time rendering on mobile devices.

**Order Independent Transparency**. Modeling transmittance remains a longstanding challenge in computer graphics, as it is essential for rendering accurate and semi-transparent structures such as flames, smoke, and clouds. Traditional methods to achieve this either successively extract depth layers, known as depth peeling (Bavoil & Myers, 2008), or store and sort fragment lists using A-buffers (Carpenter, 1984). Several approaches have proposed to circumvent explicit sorting by approximate compositing, known as Order-Independent Transparency (OIT). Similar to depth peeling, $k$-buffer methods similarly have different depth layers but store and accumulate only the first $k$ layers in a single rendering pass (Bavoil et al., 2007). Another line of approach, stochastic transparency, commonly used in Monte Carlo rendering, samples fragments based on their depth and opacity, producing visually plausible results given a sufficiently high sampling rate (Enderton et al., 2010). Recently, plenty of sort-free 3DGS works (Hou et al., 2025; Kheradmand et al., 2025; Hahlbohm et al., 2025; Sun et al., 2025) have been proposed to achieve fast rendering without sorting. The representative work is SortFreeGS (Hou et al., 2025), which enhances the rendering speed via

$$\mathbf{C} = \frac{\mathbf{c}_{bg} w_{bg} + \sum_{i=1}^{\mathcal{N}} \mathbf{c}_i \alpha_i w(d_i)}{w_{bg} + \sum_{i=1}^{\mathcal{N}} \alpha_i w(d_i)}, \quad (1)$$

where $w(d_i) = \exp\left(-\sigma_i d_i^{\beta_i}\right)$ is a weighting function. $w_{bg}$, $\sigma_i$, and $\beta_i$ represent learnable parameters, while $d_i$ denotes depth. In Eq. 1, both the numerator and denominator are summations, and since addition is commutative, these terms can be computed in any order. However, these methods cannot be directly employed in edge devices due to the large storage and significant inference delay.

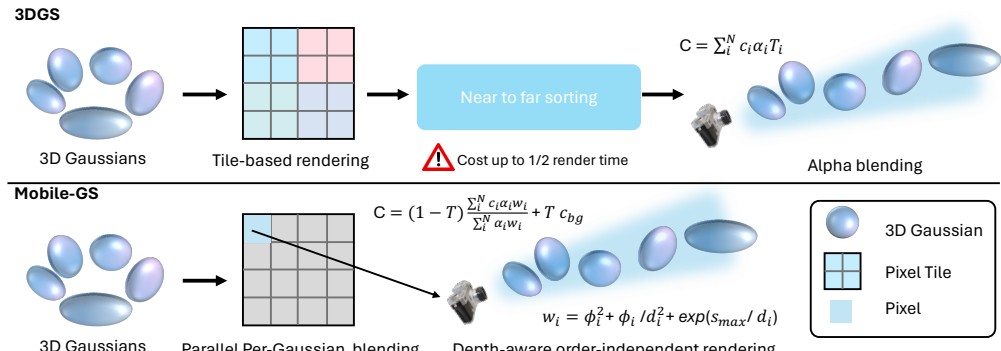

Figure 3: **Rendering pipeline of our proposed Mobile-GS compared with 3DGS.** In the inference stage, different from 3DGS, our proposed method eliminates the tile-based rendering and the 3D Gaussian sorting process typically required for accurate alpha blending. Instead, we first compute the color of each 3D Gaussian for its related pixels in parallel and accumulate the color value for each pixel. Then, we composite the foreground and background color in a single pass. To further improve performance and maintain visual quality, we propose a depth-aware order-independent rendering strategy that replaces the original sorting-dependent alpha blending.

**Gaussian Compression and Pruning**. Recent researchers have explored compressing Gaussian representations through techniques such as vector quantization (Wang et al., 2024b; Liu et al., 2024; Papantonakis et al., 2024; Xie et al., 2024) and entropy encoding (Chen et al., 2024a; Niedermayr et al., 2023). Among these, LocoGS (Shin et al., 2025) introduces a locality-aware strategy that compresses all Gaussian attributes into compact local representations. There is another line of research about Gaussian pruning technique to eliminate Gaussians. LODGE (Kulhanek et al., 2025) introduces a depth-aware 3D smoothing filter with importance-based pruning to maintain LOD visual fidelity. MaskGaussian (Liu et al., 2025) leverages a masked-rasterization technique to dynamically assess the contribution of each Gaussian and prune them with low contribution. GaussianSpa (Zhang et al., 2025) proposes a pruning technique to gradually restrict Gaussians to the target sparsity constraint and keep rendering quality. While effective in reducing redundancy, these methods often suffer from significant rendering quality degradation and still incur considerable storage overhead. To address these limitations, we propose a more efficient compression framework that preserves essential Gaussian features while achieving compact attribute representation, making it particularly suitable for deployment on resource-constrained mobile devices.

## 3 METHODOLOGY

### 3.1 DEPTH-AWARE ORDER-INDEPENDENT RENDERING

▶ **" Rendering Insight":** Order-independent rendering enables efficient Gaussian compositing.

Traditional alpha blending typically requires a depth-sorting procedure, wherein 3D Gaussians are composited in a near-to-far order to correctly accumulate color. Although this sorting-based mechanism ensures that Gaussians closer to the camera have a more significant contribution to the final image, it incurs considerable computational overhead, particularly detrimental in latency-sensitive and resource-constrained equipment like mobile devices.

To address these limitations, we propose a depth-aware order-independent rendering strategy tailored for mobile devices as illustrated in Fig. 3. Our rendering mechanism differs fundamentally from conventional alpha blending. To be specific, our rendering strategy eliminates the need for depth sorting by introducing a learnable, view-conditioned weighting scheme. The rendered pixel color $\mathbf{C}$ is computed as a weighted color accumulation from 3D Gaussians, defined as:

$$\mathbf{C} = (1 - T) \frac{\sum_{i=1}^{\mathcal{N}} \mathbf{c}_i \alpha_i w_i}{\sum_{i=1}^{\mathcal{N}} \alpha_i w_i} + T\mathbf{c}_{bg}, \tag{2}$$

where $\mathbf{c}_i$ and $\mathbf{c}_{bg}$ denote the RGB and background color of the $i$-th 3D Gaussian, respectively. $T = \prod_{j=1}^{N}(1 - \alpha_j)$ represents the global transmittance to differentiate the foreground and background.

Figure 4: **Overall illustration and visualization of view-dependent opacity modeling. Left:** We leverage an MLP fed with 3D Gaussian scale, rotation, spherical harmonics, and the vector of the camera toward the 3D Gaussian as input to predict a view-dependent opacity. **Right:** We display that our Mobile-GS removes redundant opacity and keeps important Gaussians with high opacity.

$\alpha_i = o_i \exp\left(-\frac{1}{2}\Delta x_i^T \Sigma_i^{-1} \Delta x_i\right)$ means the alpha obtained from the opacity $o_i$. $w_i$ is a depth-aware weight that modulates the contribution of each 3D Gaussian based on its scale and position related to the camera. Specifically, we utilize the inverse depth to reduce the contributions of the distant 3D Gaussians. Moreover, we increase the weight of the Gaussian with a larger scale. The weighting term $w_i$ can be determined as:

$$w_i = \phi_i^2 + \frac{\phi_i}{d_i^2} + exp(\frac{s_{max}}{d_i}), \tag{3}$$

where $d_i$ and $s_{max}$ mean the depth and the maximum scale of the $i$-th 3D Gaussian in the camera coordinate system. $\phi_i$ means the view-dependent per-Gaussian parameters, modulating the contribution of each Gaussian, which will be described in detail later. This formulation offers two key advantages. First, by removing the dependency on sorting, the proposed method enables efficient and parallel accumulation of Gaussian contributions, resulting in significantly faster rendering. It is crucial for real-time rendering on mobile hardware. Second, the weighting depth-aware modulation allows a more flexible contribution modeling, enabling 3D Gaussians to dynamically adapt to complex scene structures, without discarding information from distant Gaussians.

**Neural View-dependent Enhancement**: Although our depth-aware order-independent rendering significantly reduces computational costs, it results in a slight degradation in rendering quality. Specifically, due to the absence of strict depth-based compositing, objects that are spatially occluded or partially overlapped may exhibit undesired transparency effects. To address this issue and enhance the fidelity of the rendered images, we propose a neural view-dependent opacity enhancement strategy that incorporates explicit view-dependent information into the 3D Gaussian attributes.

As depicted in Fig. 4, we design a lightweight multi-layer perceptron (MLP) that predicts the view-dependent opacity scalar for each Gaussian. This predicted opacity aims to modulate the visibility of Gaussians in a view-aware manner, thereby compensating for the drawback of order-free rendering and improving rendering quality. The input to the MLP consists of both the geometric and appearance-related features of each 3D Gaussian. To be specific, for a given 3D Gaussian position $\mu_i$, we compute its direction from the camera center and normalize the final result to obtain the Camera-Gaussian vector $\mathbf{P}_i = \frac{\mu_i - \mathbf{t}_v}{\|\mu_i - \mathbf{t}_v\|}$, where $\mathbf{t}_v$ means the center of the $v$-th training camera. This vector encapsulates the relative orientation between the camera and the Gaussian center, providing critical view-dependent cues. To further enrich the input representation, we incorporate additional geometric and appearance-related per-Gaussian attributes, including the scale $s_i \in \mathrm{R}^3$, rotation parameter $r_i \in SO(3)$, and spherical harmonic coefficients $Y_i$, which describe the anisotropic shape and view-dependent color of the Gaussian. By feeding these composite feature vectors into the MLP, the model learns to predict a scalar that adaptively modulates the contribution of each Gaussian based on both its intrinsic attributes and the current viewing direction. Therefore, the neural enhanced view-dependent $\phi$, and opacity $o$ can be formalized as:

$$\mathbf{F} = \mathrm{MLP}_f(\mathbf{P}_i, s_i, r_i, Y_i), \ \phi_i = \mathrm{ReLU}(\mathrm{MLP}_\phi(\mathbf{F})), \ o_i = \sigma(\mathrm{MLP}_o(\mathbf{F})), \tag{4}$$

where $\sigma(\cdot)$ means the Sigmoid function. $\mathrm{MLP}_f$, $\mathrm{MLP}_\phi$, and $\mathrm{MLP}_o$ denote the MLP functions for feature $\mathbf{F}$, weight $\phi$ and opacity $o$. Specifically, view-dependent $\phi$ acts as a depth attenuation factor, adaptively scaling the influence of each Gaussian based on its distance to the camera. The view-conditioned opacity $o_i$ serves as a corrective factor in the rendering pipeline, allowing the system to dynamically suppress the transparency of the occluded regions. As a result, our Mobile-GS effectively mitigates the transparency artifacts in depth-ambiguous scenarios, leading to higher rendering quality and better preservation of scene geometry.

## 3.2 Distillation and Quantization

▶ **" Compressed Insight":** First-order SH and quantization enable efficient compression.

**First-order Spherical Harmonics Distillation:** Inspired by LightGaussian (Fan et al., 2024), it employs a distillation strategy to project third-order spherical harmonics ($3 \times 16$ coefficients) onto a second-order representation ($3 \times 9$ coefficients) for efficient rendering. Different from that, in this work, we introduce a first-order spherical harmonics ($3 \times 4$ coefficients) distillation framework that encourages a more compact model to approximate the directional radiance of a powerful teacher model via $\mathcal{L}_{\text{dstill}} = \frac{1}{|P|} \sum_{p \in P} \left\| \mathbf{C}_p^{tea} - \mathbf{C}_p \right\|$ where $P$ denotes the set of pixels. $\mathbf{C}^{tea}$ and $\mathbf{C}$ represent the rendered pixel colors from the teacher and student, respectively. Except for that, we also propose a scale-invariant depth distillation loss to impose depth supervision from the teacher model through:

$$\mathcal{L}_{\text{depth}}(\mathbf{D}, \mathbf{D}^{tea}) = \frac{1}{|P|} \sum_{p \in P} \left( \log(\hat{\mathbf{D}}_p) - \log(\hat{\mathbf{D}}_p^{tea}) \right)^2 - \frac{1}{|P|^2} \left( \sum_{p \in P} \left( \log(\hat{\mathbf{D}}_p) - \log(\hat{\mathbf{D}}_p^{tea}) \right) \right)^2,$$
(5)

where depth $\hat{\mathbf{D}}^{tea} = \mathbf{D}^{tea} + \varepsilon$ and we set $\varepsilon$ as $1e^{-8}$ for training stability. We do not use the strict restriction like L1/L2 loss since the depth from the teacher and student may have a slight difference, and the depth of the teacher model is not always reliable.

**Neural Vector Quantization:** To efficiently compress the per-Gaussian attribute vectors while preserving high rendering fidelity, we propose a neural vector quantization (NVQ) scheme tailored for 3D Gaussian splatting. Unlike traditional vector quantization methods that operate on the entire attribute vector using a single codebook, our method adopts a sub-vector decomposition strategy (Lee et al., 2025) that enhances representation flexibility and compression efficiency. Specifically, given a Gaussian attribute vector $z \in \mathbb{R}^{KL}$, we partition it into $K$ clusters $\{z_1, z_2, \ldots, z_K\}$ of length $L$ by K-Means (Hamerly & Elkan, 2003). The 3D Gaussian attributes in each cluster $z_k \in \mathbb{R}^L$ are quantized using their own codebook $\mathcal{C}^k \in \mathbb{R}^{B \times L}$ with $B$ codewords, where $B$ denotes the number of codewords per subspace. This multi-codebook quantization reduces the memory footprint of each codebook, mitigates codeword collisions, and simplifies lookup operations during inference. To further compress the final quantized attributes for storage, we apply Huffman coding to encode sequences at the end of training. This entropy-based compression technique significantly reduces the bitstream size without compromising runtime performance, enabling the deployment of our method on storage-constrained devices.

To further reduce the storage burden associated with per-Gaussian SH coefficients, we decompose the learned SH feature $Y$ into a diffuse component $h_d \in R^3$ and a view-dependent component $h_v \in R^3$ via the proposed neural vector quantization, and model them using lightweight multi-layer perceptrons (MLPs). This design eliminates the need to store high-dimensional SH coefficients directly for every Gaussian, instead leveraging compact neural functions to reconstruct SH features at inference time. The final SH features for rendering are computed as:

$$f_d = \text{MLP}_d(h_d, h_v), \ f_v = \text{MLP}_v(h_d, h_v),$$
(6)

where $\text{MLP}_d$ and $\text{MLP}_v$ are separately parameterized neural networks predicting diffuse and view-dependent spherical harmonics components, respectively. Both MLPs are quantized to 16-bit precision to minimize storage overhead while retaining representation capability. In the inference stage, we only use these MLPs to decode the SH features once as described in Eq. 6. This factorization further reduces memory costs for mobile devices.

## 3.3 Contribution-based Pruning

▶ **"Pruning Insight":** Gaussians with larger opacity and scale have more important contribution.

To reduce redundant Gaussian primitives during training, we adopt a contribution-based pruning mechanism that jointly considers opacity and spatial scale statistics. At each iteration $t$, we compute the per-primitive opacity values $o_g$ and the maximum scale $s_{\max}(g)$ across dimensions. A quantile threshold $\tau$ is applied to identify low-contributing Gaussian candidates:

$$\mathcal{C}_{\text{opacity}}^{(t)} = \{g \in \mathcal{G} \mid o_g < Q_\tau(o)\}, \ \mathcal{C}_{\text{scale}}^{(t)} = \{g \in \mathcal{G} \mid s_{\max}(g) < Q_\tau(s_{\max})\},$$
(7)

where $Q_\tau(\cdot)$ denotes the $\tau$-quantile operator, and $\mathcal{C}_{\text{prune}}^{(t)} = \mathcal{C}_{\text{opacity}}^{(t)} \cap \mathcal{C}_{\text{scale}}^{(t)}$ is the set of Gaussians selected as pruning candidates at iteration $t$. Instead of immediately removing candidates, we accumulate pruning votes for each Gaussian. Let $V_g^{(t)}$ denote the accumulated vote count for Gaussian $g$ at iteration $t$, initialized as $V_g^{(0)} = 0$. The update rule is

$$V_g^{(t+1)} = V_g^{(t)} + \mathbf{1}[g \in \mathcal{C}_{\text{prune}}^{(t)}], \ \mathcal{G}_{\text{new}} = \mathcal{G} \setminus \{g \in \mathcal{G} \mid \mathbf{1}[V_g^{(t)} > I_{prune} \cdot v]\}, \tag{8}$$

where $\mathbf{1}[\cdot]$ is the indicator function. $I_{prune}$ and $v$ are the pruning interval and a vote threshold. A Gaussian $g$ is permanently pruned if its accumulated votes exceed a threshold in every pruning interval. The updated Gaussian set is $\mathcal{G}_{\text{new}}$. This strategy mitigates noisy fluctuations in opacity or scale during early training and progressively eliminates Gaussian primitives that consistently exhibit low contribution (low opacity) and negligible geometric extent (low scale).

### 3.4 Implementation

**Training Loss**: In this work, we optimize our proposed Mobile-GS with the rendering loss same as the original 3DGS (Kerbl et al., 2023), which utilizes $\mathcal{L}_1$ and $\mathcal{L}_{\text{DSSIM}}$:

$$\mathcal{L}_{\text{rgb}} = \lambda \mathcal{L}_1(\mathbf{C}, \mathbf{C}_{\text{gt}}) + (1 - \lambda)\mathcal{L}_{\text{DSSIM}}(\mathbf{C}, \mathbf{C}_{\text{gt}}), \tag{9}$$

where $\lambda$ balance the contributions of the $\mathcal{L}_1$ and $\mathcal{L}_{DSSIM}$ loss function. It is typically set as 0.8. Therefore, the total loss can be computed via:

$$\mathcal{L} = \mathcal{L}_{\text{rgb}} + \lambda_{\text{distill}}\mathcal{L}_{\text{distill}} + \lambda_{depth}\mathcal{L}_{\text{depth}}, \tag{10}$$

where $\lambda_{\text{distill}}$ and $\lambda_{\text{depth}}$ balance the weight between the rendered image loss, the distillation loss, and the depth loss. In our experiments, we empirically set $\lambda_{\text{distill}}$ and $\lambda_{\text{depth}}$ as 1 and 0.1, respectively.

**Training Details**: We train our proposed Mobile-GS on an RTX 3090 GPU using PyTorch. We develop custom CUDA Kernels for the adaptation of our proposed depth-aware order-independent rendering. We utilize Mini-Splatting (Fang & Wang, 2024) as the teacher model in the distillation stage. The main difference between our Mobile-GS with Mini-Splatting is that it uses traditional alpha blending and does not have a quantization process. We train our method with 60k iterations. We initialize MLP$_\phi$ to output $\phi$ as 1, thereby stabilizing the training process. At the 35k-th iteration, we launch the proposed neural vector quantization as shown in Eq. 6. To improve holistic rendering performance, we adopt multi-view training (Du et al., 2026) for more powerful multi-view constraints to 3D Gaussian primitives. Additional implementation details can be found in the appendix.

**Deployment on Mobiles**: To evaluate the efficiency of our method on resource-constrained devices, we implement our approach using Vulkan 2.0, a modern, cross-platform graphics and compute API. Vulkan offers low-overhead, high-performance access to GPU hardware and is well-suited for mobile and embedded platforms due to its explicit control over rendering and memory management. This implementation enables a fair and consistent comparison of the real-time rendering performance and computational overhead on mobile GPU architectures.

## 4 Experiments

### 4.1 Quantitative and Qualitative Results

In our experiments, we compare our proposed Mobile-GS against several state-of-the-art methods, including 3DGS (Kerbl et al., 2023), LightGaussian (Fan et al., 2024), AdR-Gaussian (Wang et al., 2024a), SortFreeGS (Hou et al., 2025), Speedy-Splat (Hanson et al., 2025), C3DGS (Lee et al., 2024), GES (Ye et al., 2025), and LocoGS-S (Shin et al., 2025). As shown in Table 1, we display quantitative results of our Mobile-GS compared with these state-of-the-art methods across the three representative datasets. These methods cannot achieve high-quality novel view synthesis in terms of rendering performance and efficiency, while our proposed method can achieve comparable performance compared with 3DGS. It indicates that our Mobile-GS delivers high-quality novel view synthesis performance and provides a more flexible solution for mobile deployment. It is attributed to our proposed neural view-dependent enhancement strategy and a series of compression techniques that empower view-dependent information perception and improve rendering efficiency. In addition to the quantitative comparisons, we also present the qualitative results. As illustrated in Fig. 5, our

Table 1: **Quantitative comparisons of state-of-the-art 3D reconstruction methods on real-world datasets.** We evaluate and report performance on three commonly used datasets, such as Mip-NeRF 360 (Barron et al., 2022), Tank&Temples (Knapitsch et al., 2017), and Deep Blending (Hedman et al., 2018). We highlight the best results among the lightweight 3DGS methods.

| Dataset | Mip-NeRF360 | | | | | Tanks&Temples | | | | | Deep Blending | | | | |
| Method & Metrics | PSNR↑ | SSIM↑ | LPIPS↓ | Storage↓ | FPS↑ | PSNR↑ | SSIM↑ | LPIPS↓ | Storage↓ | FPS↑ | PSNR↑ | SSIM↑ | LPIPS↓ | Storage↓ | FPS↑ |
|---|---|---|---|---|---|---|---|---|---|---|---|---|---|---|---|
| 3DGS (Kerbl et al., 2023) | 27.21 | 0.815 | 0.214 | 839.9 MB | 174 | 23.14 | 0.841 | 0.183 | 371.5 MB | 236 | 29.41 | 0.903 | 0.243 | 697.3 MB | 214 |
| LightGaussian (Fan et al., 2024) | 27.08 | 0.801 | 0.244 | 60.4 MB | 227 | 22.61 | 0.803 | 0.264 | 29.9 MB | 392 | 28.74 | 0.856 | 0.325 | 48.2 MB | 271 |
| AdR-Gaussian (Wang et al., 2024a) | 26.95 | 0.792 | 0.259 | 358.2 MB | 254 | 22.74 | 0.809 | 0.251 | 214.6 MB | 372 | 28.92 | 0.863 | 0.305 | 251.4 MB | 284 |
| SortFreeGS (Hou et al., 2025) | 27.02 | 0.775 | 0.267 | 851.4 MB | 731 | 22.81 | 0.817 | 0.254 | 471.5 MB | 848 | 28.69 | 0.852 | 0.326 | 724.2 MB | 793 |
| Speedy-Splat (Hanson et al., 2025) | 26.92 | 0.782 | 0.296 | 79.4 MB | 401 | 23.08 | 0.821 | 0.241 | 62.4 MB | 527 | 29.11 | 0.864 | 0.309 | 71.2 MB | 463 |
| C3DGS (Lee et al., 2024) | 27.03 | 0.797 | 0.247 | 30.6 MB | 184 | **23.32** | 0.831 | 0.202 | 21.8 MB | 174 | 29.73 | 0.900 | 0.258 | 24.7 MB | 189 |
| LocoGS-S (Shin et al., 2025) | 27.02 | 0.805 | 0.241 | 8.5 MB | 292 | 23.23 | **0.837** | **0.204** | 6.8 MB | 325 | 29.76 | 0.903 | 0.251 | 7.8 MB | 322 |
| Mobile-GS (**Ours**) | **27.12** | **0.807** | **0.235** | **4.6 MB** | **1125** | 23.09 | 0.831 | 0.208 | **2.5 MB** | **1179** | **29.93** | **0.906** | **0.243** | **4.6 MB** | **1132** |

Figure 5: **Qualitative comparisons of existing methods and our proposed Mobile-GS.** We display the storage cost and FPS per scene to better demonstrate the performance of our method. We extract close-ups to highlight the differences.

Table 2: **Evaluation on the mobile device with Snapdragon 8 Gen 3 GPU.** 3DGS*, Mini-Splatting*, and SortFreeGS* mean the quantized version through Huffman encoding.

| Method | PSNR ↑ | FPS*↑ | Storage ↓ | Training ↓ |
|---|---|---|---|---|
| 3DGS* | 27.01 | 8 | 61.8 MB | 0.5 h |
| Mini-Splatting* | 27.02 | 12 | 36.9 MB | 0.4 h |
| Speedy-Splat | 26.92 | 19 | 79.5 MB | **0.4 h** |
| HAC | 26.98 | 12 | 11.8 MB | 0.7 h |
| LocoGS-S | 27.02 | 17 | 8.5 MB | 0.8 h |
| C3DGS | 27.03 | 14 | 30.6 MB | 0.6 h |
| GES | 26.98 | 18 | 29.4 MB | 0.7 h |
| SortFreeGS* | 26.74 | 24 | 64.3 MB | 1.3 h |
| Mobile-GS (**Ours**) | **27.12** | **127** | **4.6 MB** | 1.5 h |

Table 3: **Ablation Study of the proposed components.** We report results on the Mip-NeRF 360 dataset. The inference speed FPS is evaluated on the Desktop RTX 3090 GPU.

| Method | PSNR ↑ | FPS ↑ | Storage ↓ |
|---|---|---|---|
| Mobile-GS | 27.12 | 1125 | 4.6 MB |
| w/o order-independent | 27.26 | 684 | 4.5 MB |
| w/o view-dependent | 26.68 | **1227** | 4.4 MB |
| w/o neural quantization | **27.33** | 841 | 121 MB |
| w/ 0th-order SH distill. | 27.04 | 1219 | **3.6 MB** |
| w/ 2nd-order SH distill. | 27.13 | 917 | 7.3 MB |
| w/ 3rd-order SH | 27.15 | 841 | 9.6 MB |
| w/o depth in Eq.3 | 27.03 | 1167 | 4.5 MB |
| w/o scale in Eq.3 | 27.08 | 1171 | 4.5 MB |

proposed Mobile-GS can achieve sharper and more consistent novel view synthesis quality comparable to 3DGS, even better than 3DGS in the view-dependent effects. These quantitative and qualitative results demonstrate that our Mobile-GS can achieve high-quality novel view synthesis results, especially in scenes with complex geometry and lighting. This is because our proposed Mobile-GS is integrated with the proposed view-dependent enhancement to improve view-dependent rendering and facilitate the learning process of 3D Gaussians toward the complex scene structures.

**Evaluation on Mobile:** To validate the real-time performance on the edge device, we deploy our proposed Mobile-GS on a mobile device equipped with the Snapdragon 8 Gen 3 GPU for the evaluation. For a fair comparison, we also quantize 3DGS and Mini-Splatting for the deployment and evaluation. As depicted in Table 2, our proposed method outperforms these state-of-the-art methods in terms of rendering quality, speed, and storage costs,. These results demonstrate that our Mobile-GS is the most suitable for real-time rendering on mobile devices, compared with existing state-of-the-art methods. It is attributed to our proposed depth-aware order-independent rendering,

quantization, and pruning techniques, which eliminate the need of the 3D Gaussian sorting process, simultaneously render all 3D Gaussians, and significantly compress Gaussian parameters. Although our proposed Mobile-GS requires more training time, its high-quality rendering and real-time inference speed make it more suitable for mobile devices compared to existing methods.

**Runtime Analysis:** As illustrated in Fig. 6, we further provide a detailed runtime analysis of our proposed Mobile-GS evaluated on four representative scenes, covering both indoor and outdoor environments from the Mip-NeRF 360 dataset. The reported runtime accounts for all essential components involved in the rendering pipeline, including the lightweight MLPs used for view-dependent effects. Despite the inclusion of MLPs, which are often regarded as computationally demanding, our design introduces minimal overhead. This demonstrates that Mobile-GS maintains a favorable balance between computational efficiency and model performance, ensuring real-time efficiency without compromising visual fidelity.

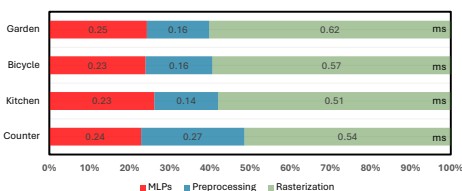

Figure 6: **Runtime analysis of Mobile-GS.**

## 4.2 ABLATION STUDY

As shown in Table 3, we conduct ablation studies to demonstrate the effectiveness of the proposed components. When we replace the proposed depth-aware order-independent rendering with the original alpha blending, although the PSNR metric is slightly improved, the rendering speed is reduced significantly. It demonstrates the critical role of the proposed order-independent rendering for better rendering efficiency. When we do not employ the proposed view-dependent enhancement, the rendered visual quality deteriorates dramatically. It is because our proposed neural view-dependent enhancement strategy can effectively mitigate the problem of depth ambiguities in overlapping geometry introduced by order-free rendering. When we do not utilize the proposed neural vector quantization, the storage cost increases dramatically, indicating its necessity for mobile deployment. When we do not leverage the proposed spherical harmonics (SH) distillation, the rendering speed is reduced, and the storage cost is increased, showing its importance for lightweight rendering. When we remove the Gaussian depth or scale in our weighting function, the rendering quality degrades, which demonstrates the significance of these two attributes for our rendering formulation. These results collectively validate the effectiveness of our components and demonstrate that each component is essential to achieving high-fidelity and real-time rendering on mobile devices.

**Neural View-dependent Enhancement:** To further demonstrate the effectiveness of the proposed neural view-dependent enhancement strategy, we present visual comparisons of its ablation as illustrated in Fig. 7. In the absence of this proposed strategy, the rendered images suffer from noticeable transparency artifacts, particularly in regions with overlapping geometry or depth ambiguity. These artifacts are primarily caused by the order-independent rendering mechanism, which does not account for the coverage of each 3D Gaussian. In contrast, when the proposed neural view-dependent enhancement strategy is incorporated, these transparency artifacts are significantly reduced. This improvement is attributed to that we use the neural network to model the relationship between the 3D Gaussian attributes and view-dependent lighting effects. This leads to more consistent and realistic rendering across viewpoints.

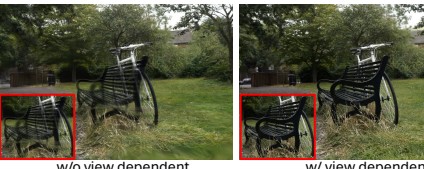

Figure 7: **Evaluation of the proposed neural view-dependent enhancement strategy.** "w/o" and "w/" mean the removal and the integration of the proposed neural view-dependent enhancement strategy.

**Contribution-based Pruning:** Our proposed contribution-based pruning strategy removes redundant Gaussian primitives by jointly considering their opacity and scale attributes. To evaluate its effectiveness, we conduct detailed ablation studies, as summarized in Table 4. The results clearly indicate that pruning based solely on either opacity or scale leads to a substantial degradation in performance, highlighting the limitation of using a single criterion. In contrast, our pruning strategy leverages the complementary nature of these two attributes: opacity reflects the visibility of a Gaussian, while scale captures its spatial influence. By integrating both factors, our method achieves a

Table 4: **Ablation study of pruning strategies.** We leverage our Mobile-GS without pruning as the baseline and leverage pruning only on opacity, scale, and both attributes.

| Method | Baseline | Opacity | Scale | Opacity & Scale |
|---|---|---|---|---|
| Num. $\times 10^6$ ↓ | 0.56 | **0.43** | 0.45 | 0.47 |
| PSNR ↑ | **27.22** | 26.84 | 26.87 | 27.12 |
| FPS* ↑ | 109 | **135** | 132 | 127 |

Table 5: **Hyperparameter analysis about the pruning threshold.** We employ our Mobile-GS without our contribution-based pruning as the baseline and adjust the pruning threshold to find a suitable trade-off.

| Threshold | Baseline | 0.1 | 0.2 | 0.4 | 0.6 |
|---|---|---|---|---|---|
| Num. $\times 10^6$ ↓ | 0.56 | 0.55 | 0.47 | 0.34 | 0.18 |
| PSNR ↑ | 27.22 | 27.15 | 27.12 | 26.47 | 25.85 |
| FPS* ↑ | 109 | 111 | 127 | 141 | 164 |

Table 6: **Adaptivity of the proposed contribution-based pruning.** Our proposed contribution-based pruning can be applied in MaksGaussian (Liu et al., 2025) and Mini-Splatting for further Gaussian pruning.

| Method | MaskGaussian | + prune. | Mini-Splatting | + prune. |
|---|---|---|---|---|
| PSNR ↑ | 27.24 | 27.16 | 27.41 | 27.38 |
| Num. $\times 10^6$ ↓ | 1.21 | 0.84 | 0.58 | 0.47 |

Table 7: **Analysis of the codebook size.** We analyze different codebook sizes on the Mip-NeRF 360 dataset to find a more balanced trade-off. The smaller codebook size means fewer storage costs.

| Codebook size | $2^6$ | $2^8$ | $2^{10}$ | $2^{12}$ |
|---|---|---|---|---|
| PSNR ↑ | 25.52 | 26.83 | 27.12 | 27.15 |
| Storage ↓ | 3.84 MB | 4.2 MB | 4.6 MB | 7.9 MB |

more balanced pruning decision, allowing us to discard a large portion of redundant primitives while maintaining high rendering fidelity. This design not only reduces memory and computational overhead but also demonstrates that effective pruning can be achieved with minimal loss of precision, thus striking a favorable trade-off between efficiency and accuracy.

In our proposed contribution-based pruning strategy, we introduce a predefined threshold $\tau$ to identify Gaussians with low contribution. A larger threshold results in more aggressive pruning and consequently fewer Gaussian points. We evaluate various threshold values, as displayed in Table 5, using Mobile-GS without pruning as the baseline and applying our contribution-based pruning strategy on top of it. We analyze thresholds ranging from 0.1 to 0.6 and observe that a threshold of 0.2 provides the best balance between rendering quality and computational efficiency. Accordingly, we adopt 0.2 as the pruning threshold in our method. As shown in Table 6, our contribution-based pruning strategy can be seamlessly integrated with various existing GS pruning methods to further reduce the number of Gaussian points. In particular, it can significantly reduce Gaussian points without substantial performance degradation.

**Neural Vector Quantization:** Our proposed neural vector quantization employs K-means clustering to encode Gaussian parameters into a compact codebook. The codebook size directly influences both rendering quality and storage cost. We evaluate a range of codebook sizes from $2^6$ to $2^{12}$, as depicted in Table 7. When the codebook is too small, the PSNR degrades drastically due to insufficient representational capacity. Conversely, an excessively large codebook requires substantially more storage. Based on this trade-off, we select a codebook size of $2^{10}$, which offers a favorable balance between lightweight storage and high-quality rendering.

## 5 CONCLUSION

In this work, we propose Mobile-GS, the first Gaussian Splatting method specifically designed for real-time rendering on mobile devices. To address the computational bottlenecks inherent in traditional 3D Gaussian Splatting, we propose a series of innovative techniques, including depth-aware order-independent rendering, neural view-dependent opacity enhancement, first-order spherical harmonics distillation, neural vector quantization, and contribution-based pruning. These innovations jointly enable Mobile-GS to achieve high-quality novel view synthesis while dramatically reducing memory, storage usage, and computational overhead. Extensive experiments demonstrate that Mobile-GS delivers rendering quality comparable to the original 3DGS, yet with a significantly smaller storage footprint and faster efficiency, achieving up to 127 FPS on the modern mobile GPU.

ACKNOWLEDGMENTS

This research is funded in part by ARC-Discovery grant (DP220100800 to XY) and ARC-DECRA grant (DE230100477 to XY). We thank all anonymous reviewers and ACs for their constructive suggestions.

REPRODUCIBILITY STATEMENT

We ensure the reproducibility of our work by detailed implementation descriptions and publicly available codes. The main paper and appendix provide implementation details. To facilitate the reproducibility, we will release all codes, including training and evaluation code, upon acceptance of the paper.

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

## A    LLM Usage Statement

In this work, we employed a Large Language Model (LLM) solely for language polishing purposes. Specifically, the LLM was used to refine grammar, improve readability, and ensure consistency in tone and style across the manuscript. Importantly, the LLM was not used for generating novel ideas, conducting analysis, or contributing to the scientific content of this work. All research design, implementation, and results presented herein are original contributions of the authors.

## B    Preliminary: 3D Gaussian Splatting

3DGS is a Gaussian-based rendering approach that leverages anisotropic 3D Gaussians to represent scenes for high-quality real-time rendering. Each 3D Gaussian $\mathcal{G}_i$ is parametrized by $(\mu_i, \Sigma_i, o_i, Y_i)$. The mean $\mu_i \in \mathbb{R}^3$ specifies the 3D position of the Gaussian in world space. The covariance matrix $\Sigma_i \in \mathbb{R}^{3\times3}$, usually symmetric and positive semi-definite, describes the anisotropic spatial extent and orientation of the Gaussian ellipsoid. $\Sigma_i$ can be decomposed into $s_i$ and $r_i$, where $s$ represents the scale of 3D Gaussians and $r$ denotes the rotation. The appearance of each Gaussian is controlled by its color $\mathbf{c}_i$ and an opacity factor $o \in [0, 1]$, where the color $\mathbf{c}$ is represented by the spherical harmonic coefficients $Y_i$.

To render an image, each Gaussian is first projected onto the image plane via a standard perspective camera model. In this process, 3DGS has a depth-sorting process to render the 3D Gaussians in order. It ensures the rendering of 3D Gaussians from near to far order. After the sorting process, the final pixel color can be computed by the alpha blending:

$$\mathbf{C} = \sum_{i=1}^{\mathcal{N}} \mathbf{c}_i \alpha_i T_i, \; T_i = \prod_{j=1}^{i-1}(1 - \alpha_j), \; \alpha_i = o_i \exp\left(-\frac{1}{2}\Delta x_i^T \Sigma_i^{-1} \Delta x_i\right), \tag{11}$$

where $T$ denotes the transmittance in the alpha blending. $\Delta x_i = x_i - \mu_i$ denotes the positional offset. $\mathcal{N}$ represents the number of 3D Gaussians. Overall, Gaussian Splatting provides a differentiable, compact, and efficient method to represent and render complex scenes, making it especially suitable for real-time applications and gradient-based optimization in neural rendering pipelines.

## C    Additional Implementation Details

**Neural View-dependent Enhancement.** To effectively capture view-dependent appearance variations, we design a lightweight three-layer multilayer perceptron $\text{MLP}_f$ with progressively decreasing neuron counts of 256, 128, and 64 per layer. This MLP extracts discriminative features from the 3D Gaussian primitives and predicts both the opacity $o$ and an auxiliary feature $\phi$. We apply a Sigmoid activation function to the opacity output to constrain it within the range $[0, 1]$, while a ReLU activation is employed for $\phi$ to enhance its representational flexibility. The network is trained jointly with the 3D Gaussian primitives for 30k iterations, ensuring that the learned features are tightly coupled with the underlying Gaussian representation.

**Neural Vector Quantization.** For the efficient compression of Gaussian attributes, we adopt a neural vector quantization scheme. Specifically, the attributes are partitioned into five groups of sub-vectors, each of which is clustered using K-means. The resulting discrete codes are then further compressed via Huffman entropy coding to reduce storage redundancy. To refine the quantized representations, we employ compact MLP modules associated with each group. These MLPs are intentionally designed with a single hidden layer consisting of 64 neurons, striking a balance between training efficiency and representational accuracy. This lightweight design significantly accelerates training while maintaining high-quality reconstruction of the Gaussian attributes.

**Distribution-based Pruning.** To remove redundant Gaussian primitives and improve efficiency, we introduce a distribution-based pruning strategy. We set the interval $I_{prune}$ to 1000 and $v$ to 0.6. Pruning is applied only during the initial 25k iterations to prevent excessive removal in later stages of training, where finer details are critical. Furthermore, we employ a redundancy identification threshold $\tau = 0.2$ to selectively discard Gaussian primitives with marginal contributions. This design allows the model to retain representation capacity while substantially reducing unnecessary primitives in early training.

Table 8: **Comparison with different sorting-free methods.** SortFreeGS* means its quantized version. We report metrics on the Mip-NeRF 360 dataset for the mobile equipped with the Snapdragon 8 Gen 3 GPU. FPS* means the rendering speed on the mobile.

| Method | Rendering | Weighting | PSNR ↑ | Storage ↓ | FPS*↑ |
|---|---|---|---|---|---|
| SortFreeGS* | $\mathbf{C} = \frac{c_{bg}w_{bg}+\sum_{i=1}^{N}c_i\alpha_i w(d_i)}{w_{bg}+\sum_{i=1}^{N}\alpha_i w(d_i)}$ | $w(d_i) = \exp\left(-\sigma d_i^{\beta}\right)$ | 26.74 | 64.3 MB | 18 |
| GES | $\mathbf{C} = \frac{C_s W_s + C_G}{W_s + W_G}$ | $W_G(\hat{\mathbf{x}}) = \sum_{i=1}^{N}[1(d_i < d_s(\hat{\mathbf{x}}) + \epsilon)]\alpha_i(\hat{\mathbf{x}})$ | 27.02 | 29.4 MB | 24 |
| **Ours** | $\mathbf{C} = (1-T)\frac{\sum_{i=1}^{N}\mathbf{c}_i\alpha_i w_i}{\sum_{i=1}^{N}\alpha_i w_i} + T\mathbf{c}_{bg}$ | $w_i = \phi_i^2 + \frac{\phi_i}{d_i^2} + exp(\frac{s_{max}}{d_i})$ | **27.12** | **4.6** MB | **127** |

# D   DISCUSSION AND LIMITATIONS

## D.1   DIFFERENCE WITH SORTING-FREE METHODS

Although SortFreeGS (Hou et al., 2025), GES (Ye et al., 2025), and our proposed Mobile-GS employ sorting-free rendering, Mobile-GS serves as a more comprehensive and efficient rendering method as depicted in Table 8. It is worth noting that SortFreeGS* refers to the quantized version of SortFreeGS, as the original method does not include a quantization stage. In terms of PSNR, storage cost, and rendering FPS, our Mobile-GS consistently achieves superior performance over prior sorting-free techniques. This improvement stems from our integrated design that incorporates quantization, pruning, and a view-dependent enhancement mechanism. With respect to rendering formulations, GES follows a formulation similar to SortFreeGS, whereas our method adopts a transmittance proxy enriched with view-dependent modulation to more effectively capture the underlying 3D scene structure.

The Gaussian weight computation also differs substantially across these approaches. SortFreeGS leverages the Gaussian depth to modulate its contribution but does not account for the Gaussian scale, which we find to be critical. GES, on the other hand, relies on a two-stage rendering. It first renders a depth image using conventional volume rendering and then filters out distant Gaussians by comparing their depths against the rendered depth map for later sorting-free rendering. This two-stage rendering pipeline relies on precise depth rendering and increases computational load, so it is not well-suited for mobile deployment. In contrast, Mobile-GS exploits both depth and scale attributes of each Gaussian to compute an importance weight, reflecting the intuition that farther Gaussians should have lower contribution, while larger Gaussians typically provide more meaningful rendering evidence.

Theoretically, A key challenge for sorting-free methods is the potential order ambiguity in regions where geometry overlaps. SortFreeGS attempts to address this by introducing additional spherical harmonics parameters to model view-dependent opacity. However, this design incurs significant overhead and is unfavorable for practical mobile usage. Our Mobile-GS resolves this limitation by enhancing the view-dependent effect through a learnable parameter $\phi$, predicted by a lightweight MLP conditioned on Gaussian attributes. This formulation achieves high-quality rendering without introducing a prohibitive computational or memory burden. Overall, Mobile-GS is carefully tailored to minimize resource consumption, reduce Gaussian parameter storage, and maintain real-time rendering performance on mobile hardware.

## D.2   DISCUSSION

Mobile-GS is a Gaussian-based method that can achieve real-time rendering on mobile and resource-constrained platforms without significantly sacrificing rendering quality. The proposed depth-aware order-independent rendering replaces traditional alpha blending with a sorting-free scheme, substantially improving runtime efficiency. Combined with neural view-dependent enhancements and spherical harmonics distillation, our approach maintains visual fidelity even under complex scenes. To address memory limitations, a neural vector quantization strategy is employed, improving storage efficiency and enabling large-scale scene representations to be deployed on mobile devices with limited memory.

Experimental results demonstrate that Mobile-GS achieves a compelling balance among rendering speed, storage footprint, and visual quality. It consistently outperforms existing lightweight Gaus-

Table 9: **Per-scene PSNR results of state-of-the-art novel view synthesis methods on Mip-NeRF 360 dataset (Barron et al., 2022)**. The best results are highlighted.

| Method | bicycle | garden | stump | flowers | treehill | counter | kitchen | room | bonsai |
|---|---|---|---|---|---|---|---|---|---|
| 3DGS (Kerbl et al., 2023) | **25.23** | **27.38** | 26.55 | **21.44** | 22.49 | 28.70 | 30.32 | 30.63 | **31.98** |
| Speedy-Splat (Hanson et al., 2025) | 24.78 | 26.70 | 26.79 | 21.21 | 22.57 | 28.28 | 29.91 | **30.99** | 31.29 |
| Mobile-GS (**Ours**) | 24.91 | 26.65 | **26.82** | 21.41 | **22.77** | 28.82 | 30.47 | 30.95 | 31.25 |

Table 10: **Per-scene SSIM results of state-of-the-art novel view synthesis methods on Mip-NeRF 360 dataset (Barron et al., 2022)**. The best results are highlighted.

| Method | bicycle | garden | stump | flowers | treehill | counter | kitchen | room | bonsai |
|---|---|---|---|---|---|---|---|---|---|
| 3DGS (Kerbl et al., 2023) | **0.765** | **0.864** | 0.770 | **0.602** | 0.633 | **0.907** | **0.925** | 0.918 | **0.940** |
| Speedy-Splat (Hanson et al., 2025) | 0.704 | 0.815 | 0.765 | 0.561 | 0.590 | 0.878 | 0.894 | 0.905 | 0.927 |
| Mobile-GS (**Ours**) | 0.740 | 0.823 | **0.777** | 0.593 | **0.643** | 0.905 | 0.920 | **0.924** | 0.936 |

Table 11: **Per-scene LPIPS results of state-of-the-art novel view synthesis methods on Mip-NeRF 360 dataset (Barron et al., 2022)**. The best results are highlighted.

| Method | bicycle | garden | stump | flowers | treehill | counter | kitchen | room | bonsai |
|---|---|---|---|---|---|---|---|---|---|
| 3DGS (Kerbl et al., 2023) | **0.211** | **0.108** | **0.217** | **0.339** | **0.329** | 0.201 | **0.127** | 0.220 | 0.205 |
| Speedy-Splat (Hanson et al., 2025) | 0.333 | 0.213 | 0.288 | 0.419 | 0.463 | 0.260 | 0.198 | 0.260 | 0.231 |
| Mobile-GS (**Ours**) | 0.270 | 0.180 | 0.250 | 0.356 | 0.354 | **0.195** | 0.132 | **0.194** | **0.187** |

sian Splatting methods across multiple benchmarks, highlighting the effectiveness of our proposed components, including depth-aware order-independent rendering, neural view-dependent enhancement, spherical harmonics distillation, neural vector quantization, and contribution-based pruning.

## D.3 LIMITATIONS

Despite its advantages, Mobile-GS contains several limitations: (1) Training Cost and Complexity: Although inference is fast, training Mobile-GS remains computationally intensive due to the proposed components (*e.g.*, spherical harmonics distillation, neural vector quantization, neural view-dependent enhancement). Additionally, the model requires pretraining on desktop GPUs before mobile deployment, limiting its accessibility for real-time data acquisition and retraining on the device. (2) Scene Generalization: While Mobile-GS performs well on standard benchmarks, it is optimized per-scene and does not generalize across scenes without retraining. This limits its immediate usage in applications requiring dynamic scene capture or rendering in unseen environments, such as real-time AR reconstruction. (3) Quantization Degradation: Although the proposed neural vector quantization is highly effective in compressing Gaussian attributes, there still remains a trade-off between compression ratio and reconstruction quality, especially for fine-grained appearance details. Excessive quantization may introduce minor color shifts or blurring artifacts in highly textured regions.

## E ADDITIONAL QUALITATIVE AND QUANTITATIVE RESULTS

As shown in Table 9, 10 and 11, we provide detailed per-scene quantitative results, between our proposed Mobile-GS, 3DGS (Kerbl et al., 2023), SortFreeGS (Hou et al., 2025), and Speedy-Splat (Hanson et al., 2025). The results indicate that Mobile-GS achieves performance comparable to these state-of-the-art methods and even surpasses them in several scenes. Additional qualitative results of novel view synthesis are presented in Fig. 8, where Mobile-GS demonstrates superior rendering quality. This improvement is primarily attributed to our proposed neural view-dependent enhancement strategy, which facilitates better fitting and adaptation of 3D Gaussians toward view-dependent effects. It enhances the representation capacity of 3D Gaussian properties, thus achieving better results. These results further demonstrate that our proposed Mobile-GS not only supports real-time rendering on mobile devices but also maintains high-quality novel view synthesis.

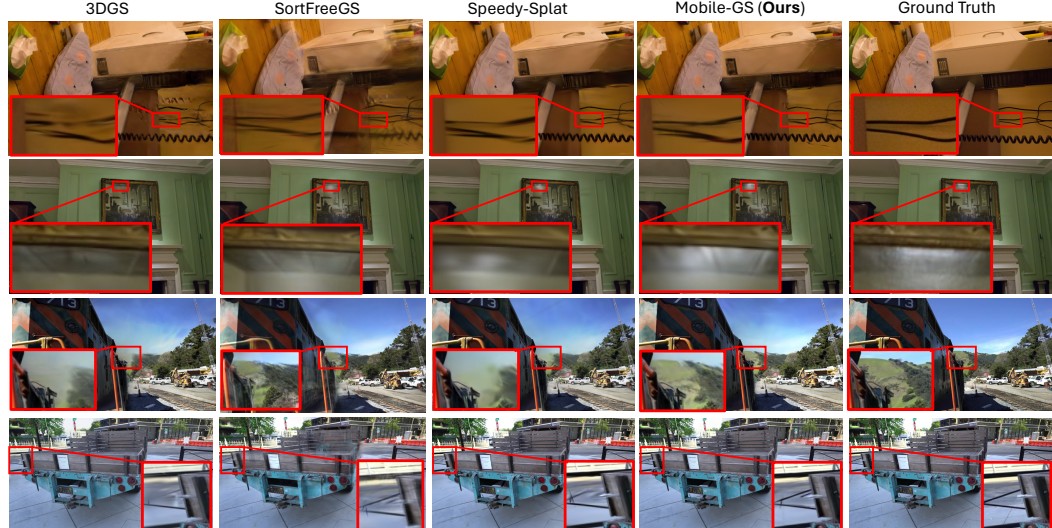

Figure 8: **Additional visual comparisons of Speedy-Splat, SortFreeGS, 3DGS, and our proposed Mobile-GS.** We highlight the close-up for better differentiation.

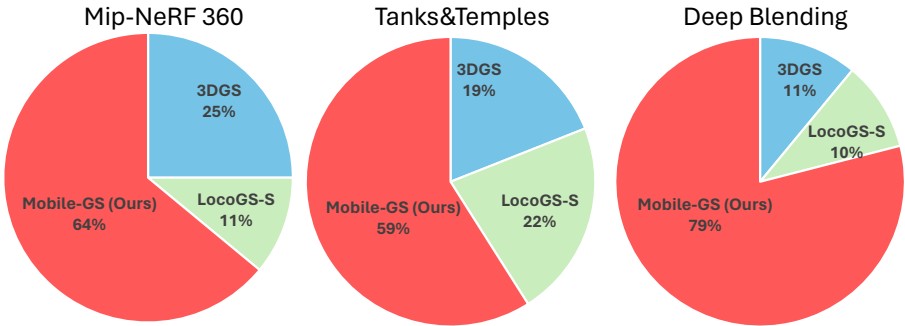

Figure 9: **User study of rendering quality between our proposed Mobile-GS, 3DGS (Kerbl et al., 2023), and LocoGS-S (Shin et al., 2025).** The higher rate means more users like it.

## F  USRER STUDY

As illustrated in Fig. 9, we conduct a user study to evaluate the rendering quality of our proposed Mobile-GS. To be specific, we train and render on the publicly available datasets, including Mip-NeRF 360 (Barron et al., 2022), Tank&Temples (Knapitsch et al., 2017), and Deep Blending (Hedman et al., 2018), with 3DGS (Kerbl et al., 2023), LocoGS-S (Shin et al., 2025), and our proposed Mobile-GS. For a fair comparison, we also quantize 3DGS. A total of 30 volunteers participated in the study, rating the quality of videos involving novel view synthesis produced by each method. These results suggest that most of the participants preferred the renderings produced by our Mobile-GS, indicating higher visual quality. This preference is largely attributed to our tailored design for mobile platforms, incorporating neural view-dependent enhancement and neural vector quantization, while 3DGS often exhibits floaters and rendering artifacts. These results further demonstrate that Mobile-GS delivers visually appealing rendering results and performance, especially under resource-constrained mobile environments.

## G  ADDITIONAL MOBILE TESTING

For a more comprehensive analysis, we further conduct detailed evaluations on a mobile device equipped with a Snapdragon 8 Gen 3 GPU. As summarized in Table 12, we report both the cold-start FPS (measured immediately at launch) and the steady-state FPS (measured after thermal equi-

Table 12: **Steady-state FPS evaluation on the mobile.** The larger FPS means faster rendering speed.

| Method | 3DGS | Speedy-Splat | SortFreeGS | Mobile-GS (Ours) |
|---|---|---|---|---|
| Cold-start FPS ↑ | 8 | 19 | 24 | **127** |
| Steady-state FPS ↑ | 3 | 10 | 18 | **74** |

Table 13: **Power draw measurement on the mobile.** We measure on different Vulkan operators and report their power (W) on the mobile with the Snapdragon 8 Gen 3 GPU.

| Method | Preprocessing | Sorting | MLP | Rasterization | Total |
|---|---|---|---|---|---|
| 3DGS* | 1.64 | 2.09 | 0 | 2.16 | 5.89 |
| SortFreeGS* | 1.78 | 0 | 0 | 2.25 | 4.03 |
| Mobile-GS | 0.17 | 0 | 0.24 | 0.42 | 0.83 |

librium). This allows us to clearly analyze the performance change over time and the impact of thermal throttling on mobile rendering. On mobiles, FPS drops over time because of thermal throttling, power limits, GPU clock downscaling, and NPU/CPU frequency limits. We can find that our Mobile-GS can still achieve 74 Steady-state FPS, which demonstrates the effectiveness of our Mobile-GS for real-time rendering on mobiles.

We further report the power-consumption measurements on the mobile device, as summarized in Table 13. Using the Qualcomm Trepn Profiler, we measure the power draw (W) of the Vulkan operators and MLP when running on the Mip-NeRF 360 dataset, and compare our Mobile-GS with 3DGS* and SortFreeGS*. Here, 3DGS* and SortFreeGS* denote their quantized variants adapted for deployment on a Snapdragon 8 Gen 3 mobile GPU. The results show that Mobile-GS achieves the lowest power consumption among all methods, highlighting the practical efficiency and suitability of our approach for mobile deployment.

