# OpenReview forum: "Mobile-GS: Real-time Gaussian Splatting for Mobile Devices"
_ICLR.cc/2026/Conference — ICLR 2026 Poster_

### Official Review · Reviewer_JkAA · 2025-10-26

**Soundness:** 3
**Presentation:** 3
**Contribution:** 3
**Rating:** 4
**Confidence:** 4

**Summary:**

This paper aims to run 3D Gaussian Splatting representations on mobile devices. To this end, the authors first identified that sorting consumes a large portion of the computational cost. Therefore, the paper proposes a sort-free algorithm with lower computational costs compared to other existing sort-free methods. This is combined with view-dependent features to mitigate artifacts. To make the representation even more lightweight, the authors introduce color distillation, vector quantization, and pruning. The experiments cover both desktop GPUs and mobile devices, evaluated with both numerical metrics and user preference studies. With all these components, the results suggest that the proposed method increases FPS and decreases storage and peak memory, all while maintaining representation quality.

**Strengths:**

- Strong performance. The proposed method shows strong performance over baselines and is well-evaluated on mobile devices. In addition, it provides thorough ablation studies that help the reader understand the role and effect of each component.
- The paper provides a runtime analysis that identifies how expensive certain operations are, which operations are bottlenecks, and how these can be reduced for mobile devices. It also shows the resulting share of operations after removing sorting.
- Beyond numerical metrics, the inclusion of a user study strengthens the performance claims from a user's perspective.

**Weaknesses:**

- The sort-free method seems to be a core part of this paper's novelty. However, a critical equation for the weighting term (Eq. 3) is not supported by or provided with an ample description of its underlying idea or theoretical grounding. The paper would also benefit from a direct comparison of the equations from different sort-free methods. Although there are descriptive paragraphs in the supplementary, the relationship between existing methods is not clearly shown at the equation level.
- Although distillation is an important part of this method, the necessity of a teacher model is not fully justified, nor is the setup adequately explained. It's unclear why this method requires distillation unlike other methods. Since it requires much longer training, it would be helpful if the paper explained why the current comparison (directly comparing "from scratch" methods with the proposed method that additionally requires a teacher model) is fair.

**Questions:**

- **Weighting term (Eq. 3)**: Could the authors provide the theoretical motivation or intuition behind the weighting term in Eq. 3? Including comparisons at the equation level would help readers better understand the context and the novelty of this component.
- **Necessity of distillation**: Is the distillation step truly necessary for achieving strong performance? Since the proposed depth-aware, order-independent rendering pipeline could, in principle, be trained from scratch, it is unclear why the method depends on distillation rather than following the same training approach as the baselines.

If these points are adequately addressed, I would be willing to raise my score.

---

> ### Author Response · Authors · 2025-11-21
> **Response to Reviewer JkAA (Part 1)**
>
> We sincerely appreciate your effort and time for reviewing our work.  We are grateful for your recognition of the performance of our proposed method and the comprehensiveness of the experiments. Please find our point-to-point response to the review comments.
>
> **Q1: Discussion with other sorting-free methods at the equation level.**
>
> A1: Thank you for your insightful suggestion. Compared to existing sorting-free methods, Mobile-GS offers a more comprehensive and mobile-oriented design, enabling **high-quality** and **real-time** rendering on resource-constrained devices. A summary of these comparisons on the Mip-NeRF 360 dataset is provided in the Table below.
>
> | Method | Rendering | Weighting | PSNR$\uparrow$ | Storage$\downarrow$ | FPS$\uparrow$ | How to solve order-ambigous problem |
> | --- | --- | --- | --- | --- | --- | --- |
> | SortFreeGS* | $ \mathbf{C} = \frac{c_{bg} w_{bg} + \sum_{i=1}^{\mathcal{N}} c_i \alpha_i w(d_i) }{ w_{bg} + \sum_{i=1}^{\mathcal{N}} \alpha_i w(d_i)}$ | $w(d_i) = \exp\left(-\sigma d_i^\beta\right)$ | 26.74 | 64.3 MB | 18 | Introducing addtional Sheprical Harmonics (SH) parameters |
> | GES | $ \mathbf{C} = \frac{C_sW_s+C_G}{W_s+W_G}$ | $ W_G(\hat{\mathbf{x}})=\sum_{i=1}^N[1(d_i<d_s(\hat{\mathbf{x}})+\epsilon)]\alpha_i(\hat{\mathbf{x}})$ | 27.02 | 29.4 MB | 24 | Using two-stage rendering(1. volume rendering for depth filtering farther Gaussians, 2. rendering remained Gaussians with order-independet rendering) |
> | Ours | $\mathbf{C}=(1-T)\frac{\sum_{i=1}^{\mathcal{N}} c_i \alpha_i w_i} {\sum_{i=1}^{\mathcal{N}} \alpha_{i}w_i  } +T\mathbf{c}_{bg}$ | $ w_i= \phi^2_i + \frac{\phi_i }{d_i^2} + exp(\frac{s_{max}}{d_i})$ | 27.12 | 4.6 MB | 127 | Introducing a lightweight MLP conditioned on Gaussian-camera vetors to enhance 3D spatial understanding ability |
>
> Here, SortFreeGS* refers to its quantized variant, as the original SortFreeGS [1] does not include quantization. In terms of PSNR, storage cost, and FPS, Mobile-GS consistently outperforms previous sorting-free approaches such as SortFreeGS [1] and GES [2]. The advantages of our Mobile-GS stem from the integration of our proposed neural vector quantization, contribution-based pruning, and view-dependent enhancement strategy.
>
> Regarding rendering formulations, GES follows a similar scheme to SortFreeGS, whereas Mobile-GS introduces a transmittance proxy along with view-dependent enhancements to more effectively understand 3D spatial and structure information. Since these methods composite Gaussians in an order-independent way, there must be an order-ambiguous problem. To solve this problem, SortFreeGS employs additional spherical harmonics parameters to model view-dependent opacity, which significantly increases storage requirements and limits its practicality on mobile hardware. In contrast, our Mobile-GS leverages a lightweight MLP conditioned on Gaussian-camera vectors and other Gaussian attributes to predict view-dependent opacity in a efficient way.
>
> In terms of Gaussian weight computation, SortFreeGS modulates weights based on Gaussian depth and two learnable parameters while neglecting the Gaussian scale. GES relies on a two-stage process: first rendering a depth image via standard volume rendering to filter distant Gaussians using this depth map, then rendering with remained Gaussian in a sorting-free way. This two-stage rendering is computationally expensive and unsuitable for mobile deployment. In contrast, Mobile-GS simultaneously leverages Gaussian depth $d_i$ and scale $s_{max}$ to modulate weights and incorporates a view-dependent parameter $\phi$ to further enhance rendering quality with a lightweight MLP. Specifically, the inverse depth is used to reduce the influence of farther Gaussians, while larger Gaussians are given greater contribution during rendering. $\phi$ is a view-dependent parameter predicted by the lightweight MLP conditioned on viewing direction for view-dependent enhancement. We set it as quadratic to enhance its non-linear mapping capability.
>
> Overall, Mobile-GS achieves **lower computational cost**, **higher rendering quality**, and **real-time performance** on mobile devices through a dedicated and highly efficient system design. These improvements demonstrate that our method not only maintains satisfactory rendering quality but also ensures practical applicability in mobile and resource-limited environments.
>
> [1] Hou Q, Rauwendaal R, Li Z, et al. Sort-free gaussian splatting via weighted sum rendering[J]. arXiv preprint arXiv:2410.18931, 2024.
>
> [2] Ye K, Shao T, Zhou K. When gaussian meets surfel: Ultra-fast high-fidelity radiance field rendering[J]. ACM Transactions on Graphics (TOG), 2025, 44(4): 1-15.

---

> ### Author Response · Authors · 2025-11-21
> **Response to Reviewer JkAA (Part 2)**
>
> **Q2: The necessity of distillation.**
>
> A2: Thank you for your valuable insight. Our proposed Mobile-GS incorporates a spherical harmonics (SH) distillation process to reduce the number of SH parameters while preserving high rendering quality. To investigate the necessity of distillations, we conducted a series of experiments of Mobile-GS variants on the Mip-NeRF 360 dataset, as summarized in the Table below.
>
> | Method | Mobile-GS (1-order SH distillation) | 0th-order SH distillation | 2nd-order SH distillation | 3rd-order SH( train from scratch) | 1st-order SH (train from scratch) |
> | --- | --- | --- | --- | --- | --- |
> | PSNR $\uparrow$ | 27.12 | 27.04 | 27.13 | 27.15 | 26.41 |
> | FPS $\uparrow$ | 1125 | 1219 | 917 | 841 | 1125 |
> | Storage $\downarrow$ | 4.6 MB | 3.6 MB | 7.3 MB | 9.6 MB | 4.6 MB |
>
> The results show that, with the proposed SH distillation, 1st-order SH parameters achieve rendering performance close to that of 3rd-order SH, demonstrating the effectiveness of our distillation strategy. However, further reducing the SH order to 0th-order leads to noticeable degradation in rendering quality. If we do not use the proposed distillation and directly train from scratch with 1st-order SH, the rendering performance significantly drops.
>
> Based on these findings, we adopt 1st-order SH distillation in our Mobile-GS implementation, balancing high-quality rendering with fast inference speed for real-time applications on mobile devices.
>
> **Q3: The motivation and intuition behind the weighting term in Eq. 3.**
>
> A3: Traditional 3D Gaussian Splatting (3DGS) relies on **depth sorting** to blend Gaussians from near to far, ensuring correct compositing. However, this sorting step is the **major computational bottleneck** (see Fig. 2 of the paper).
> On mobile GPUs (e.g., Snapdragon 8 Gen 3), sorting tens of thousands of Gaussians per frame is infeasible. Eq. 3 is designed to avoid depth sorting and maintain comparable rendering quality of the sorted rendering methods, enabling real-time and high-quality inference.
>
> The motivation for Eq. 3 is therefore to design a **mathematically order-independent** weighting scheme that can *approximate the depth compositing effect* without explicit sorting. To mimic the physical intuition behind alpha blending (where closer and larger Gaussians contribute more to the pixel color), the weighting term $ w_i= \phi^2_i + \frac{\phi_i }{d_i^2} + exp(\frac{s_{max}}{d_i}) ,$ is designed to:
>
> - **Suppress far-away Gaussians**: The term $\frac{\phi_i }{d_i^2}$ decreases the weight quadratically with depth $d_i^2$, ensuring that distant Gaussians contribute less.
> - **Emphasize larger Gaussians**: The exponential term $exp(\frac{s_{max}}{d_i})$ strengthens contributions from large-scale Gaussians that occupy a bigger image region.
> - **Maintain view-dependent effect**: The learnable per-Gaussian factor $\phi$ predicted from an MLP accounts for view-dependent effects such as specular, transparent effects, and lighting variations.
>
> These components combine together, approximating the depth-sorting rendering in a continuous and differentiable form, enabling **order-independent compositing for real-time rendering on mobile platforms**.
>
> **Q4: Why not directly pretrain on Mobile-GS**
>
> A4: Thank you for your insightful feedback. In our proposed 1st-order SH distillation, we leverage a 3rd-order SH model to guide the training of a 1st-order SH model. Specifically, we first train  Mini-Splatting and adopt it as the teacher model to reduce overall training costs. In response to Q2, we have demonstrated the necessity of our 1st-order SH distillation.
>
> While it is possible to use the teacher Mobile-GS to distill the student Mobile-GS, this approach incurs significantly higher training costs, as shown in the Table below. The increased cost arises from the inclusion of an MLP in Mobile-GS, which is used to enhance the view-dependent effects but also increases computational and memory requirements during training. It is because the input of this MLP is the whole Gaussian attributes, and it predicts view-dependent parameters for each Gaussian. By contrast, selecting Mini-Splatting as the teacher model enables more efficient distillation without sacrificing the performance of the resulting student model. This design choice strikes a balance between high rendering quality and practical training efficiency.
>
> | Method | Mobile-GS (Mini-Splatting distill) | Mobile-GS (self-distill) |
> | --- | --- | --- |
> | PSNR $\uparrow$ | 27.12 | 27.14 |
> | Training Time $\downarrow$ | 1.5 h | 2.2 h |

---

### Official Review · Reviewer_xVP3 · 2025-10-29

**Soundness:** 3
**Presentation:** 3
**Contribution:** 3
**Rating:** 6
**Confidence:** 4

**Summary:**

This paper proposes a novel method called Mobile-GS, which enables the deployment of the Gaussian Splatting (GS) method on mobile devices and achieves real-time rendering performance. The authors analyze the limitations of GS rendering and identify alpha blending as the main bottleneck. To address this issue, they propose a depth-aware, order-independent rendering scheme that eliminates the need for sorting. In addition, they introduce a neural view-dependent enhancement strategy to mitigate rendering artifacts. Post-processing techniques such as distillation and quantization are also employed to achieve efficient rendering on mobile devices.

**Strengths:**

1. The authors provide deep insights into the rendering strategy of 3DGS and effectively analyze its bottlenecks.
2. The proposed depth-aware order-independent rendering approach is interesting and appears novel.
3. The use of quantization and pruning methods successfully enables the deployment of 3DGS on mobile devices.

**Weaknesses:**

1. Do the authors analyze the precision or deviation of the depth-aware rendering theoretically? It would be helpful if they could provide a mathematical analysis or proof, along with additional exploratory experiments.
2. The reviewer would like to know whether neural vector quantization and pruning affect rendering quality. Could the authors provide evaluation results rendered on mobile devices and compare them with other methods?
3. What is the time cost of the proposed approach? It would strengthen the paper if the authors could provide this metric and compare it with existing methods.

**Questions:**

Please refer to Weaknesses.

---

> ### Author Response · Authors · 2025-11-21
> **Response to Reviewer xVP3 (Part 1)**
>
> Thanks for your effort invested in reviewing our work. We are grateful for your recognition to the novelty of our proposed method.  In response to your questions, please find our clarifications below.
>
> **Q1: Motivation for the depth-aware rendering and additional ablation studies**
>
> A1: Thank you for your valuable comment. We empirically design our proposed depth-aware rendering Eq.3: $w_i= \phi^2_i + \frac{\phi_i }{d_i^2} + exp(\frac{s_{max}}{d_i})$.
>
> Since our rendering process does not include the original sorting step, the Gaussian compositing becomes order-ambiguous. To mitigate this issue, we incorporate inverse depth into the weighting function, which effectively reduces the contributions of Gaussians that are farther from the camera. Intuitively, Gaussians with larger scales are considered to have a more significant contribution to the appearance and geometry of the 3D scene during rendering, as they contribute broader spatial influence. Furthermore, the parameter $\phi$ plays a critical role in modeling view-dependent effects. Specifically, $\phi$ is predicted using a multi-layer perceptron (MLP) conditioned on view-dependent information, as the Gaussian-camera vector. This design allows the rendering to adapt dynamically to different viewpoints, enhancing visual fidelity. An ablation study demonstrating the impact of this view-dependent enhancement is provided in Figure 7.
>
> We also conducted ablation studies on the Mip-NeRF 360 dataset to validate the effects of removing depth $d_i$ and scale $s_{max}$ from the weighting formulation. The results, summarized below, demonstrate that both attributes are essential for the rendering mechanism. Removing either parameter leads to a noticeable degradation in performance. This can be attributed to the fact that depth and scale directly influence the weight calculation: far-away Gaussians contribute less, and larger Gaussians contribute more prominently. By incorporating these attributes, our method is able to more accurately estimate the contribution of each Gaussian, resulting in improved rendering quality and a more precise representation of 3D scenes.
>
> | Method | Mobile-GS | w/o $d_i$ in Eq.3  | w/o $s_{max}$ in Eq.3  |
> | --- | --- | --- | --- |
> | PSNR $\uparrow$ | 27.12 | 27.03 | 27.08 |

---

> > ### Author Response · Authors · 2025-11-21
> > **Response to Reviewer xVP3 (Part 2)**
> >
> > **Q2: Whether neural vector quantization and pruning affect rendering quality and compare with other methods.**
> >
> > A2: Thank you for your insightful feedback. Our Mobile-GS leverages the proposed neural vector quantization and contribution-based pruning to reduce storage costs. We conducted ablation studies on the Mip-NeRF 360 dataset to evaluate the effects of removing these components, as summarized in the Table below. The neural vector quantization and pruning result in a minor PSNR reduction of approximately 0.1–0.2 dB, which is acceptable. However, without neural vector quantization, the storage cost increases substantially to 121 MB, leading to the approach being impractical for mobile deployment. Meanwhile, without the proposed contribution-based pruning, it increases the storage footprint from 4.6 MB to 6.8 MB. To enable lightweight deployment, we choose to incorporate these two components.
> >
> > | Method | Mobile-GS | w/o neural vector quantization | w/o contirbution-based pruning |
> > | --- | --- | --- | --- |
> > | PSNR $\uparrow$ | 27.12 | 27.33 | 27.22 |
> > | Storage $\downarrow$ | 4.6 MB | 121 MB | 6.8 MB |
> >
> > To evaluate the effectiveness of the proposed neural vector quantization, we conducted an experiment on the Mip-NeRF 360 dataset comparing Mobile-GS with traditional vector quantization. As shown in the Table below, Mobile-GS equipped with our proposed neural vector quantization achieves higher PSNR while requiring lower storage. This improvement is because our proposed neural vector quantization effectively clusters subvectors of Gaussian attributes and employs MLPs to encode SH parameters, thereby reducing storage without causing significant performance degradation.
> >
> > | Method | Mobile-GS (Neural Vector Quantization) | Mobile-GS (Vector Quantization) |
> > | --- | --- | --- |
> > | PSNR $\uparrow$ | 27.12 | 26.98 |
> > | Storage $\downarrow$ | 4.6 MB | 6.3 MB |
> >
> > Our proposed contribution-based pruning is a plug-and-play technique that can be integrated into existing Gaussian pruning methods to further eliminate redundant Gaussians. As shown in the Table below, MaskGaussian and Mini-Splatting are both highly effective GS variants for reconstructing 3D scenes with a limited number of Gaussians. By incorporating our contribution-based pruning into these methods, the 3D Gaussians can be further refined, resulting in a more compact representation with fewer Gaussian points and only incurring a minor PSNR reduction. These results demonstrate the effectiveness of our contribution-based pruning for precise and efficient Gaussian pruning.
> >
> > | Method | MaskGaussian [1] | MaskGaussian + Our pruning | Mini-Splatting | Mini-Splatting + Our pruning |
> > | --- | --- | --- | --- | --- |
> > | PSNR $\uparrow$ | 27.24 | 27.16 | 27.41 | 27.38 |
> > | Gaussian Number $\times 10^6$ $\downarrow$ | 1.21 | 0.84 | 0.58 | 0.47 |
> >
> > [1] Liu Y, Zhong Z, Zhan Y, et al. Maskgaussian: Adaptive 3d gaussian representation from probabilistic masks[C]//Proceedings of the Computer Vision and Pattern Recognition Conference. 2025: 681-690.

---

> ### Author Response · Authors · 2025-11-21
> **Response to Reviewer xVP3 (Part 3)**
>
> **Q3: The time cost of the proposed approach and compare it with existing methods.**
>
> A3: Thank you for your insightful suggestion.  We have added the training time metric on the Mip-NeRF 360 dataset on the mobile platform to Table 2 in the revised version, as shown below.
>
> | Method | PSNR $\uparrow$ | FPS* $\uparrow$ | Storage $\downarrow$ | Peak Memory $\downarrow$ | Training $\downarrow$ |
> | --- | --- | --- | --- | --- | --- |
> | 3DGS* | 27.01 | 8 | 61.8 MB | 9489 MB | 0.5 h |
> | Mini-Splatting* | 27.02 | 12 | 36.9 MB | 5841 MB | 0.4 h |
> | Speedy-Splat | 26.92 | 19 | 79.5 MB | 6647 MB | 0.4 h |
> | HAC | 26.98 | 12 | 11.8 MB | 7591 MB | 0.7 h |
> | LocoGS-S | 27.02 | 17 | 8.5 MB | 7018 MB | 0.8 h |
> | C3DGS | 27.03 | 14 | 30.6 MB | 7423 MB | 0.6 h |
> | GES | 26.98 | 18 | 29.4 MB | 7146 MB | 0.7 h |
> | SortFreeGS* | 26.74 | 24 | 64.3 MB | 9628 MB | 1.3 h |
> | Mobile-GS  | 27.12 | 127 | 4.6 MB | 4865 MB | 1.5 h |
>
> Our proposed method incurs additional training time due to a pretraining stage for distillation. This distillation process enables the compression of 3rd-order SH parameters into 1st-order SH parameters while maintaining high rendering quality. To illustrate the importance of pretraining and distillation, we conducted ablation studies on the Mip-NeRF 360 dataset, as summarized in the Table below.
>
> | Method | Mobile-GS (1-order SH distillation) | 2-order SH distillation | 3-order SH | 1-order SH w/o pretraining |
> | --- | --- | --- | --- | --- |
> | PSNR $\uparrow$ | 27.12 | 27.13 | 27.15 | 26.41 |
> | FPS $\uparrow$ | 1125 | 917 | 841 | 1125 |
> | Storage $\downarrow$ | 4.6 MB | 7.3 MB | 9.6 MB | 4.6 MB |
>
> The results show that Mobile-GS with the proposed 1st-order SH distillation achieves PSNR performance comparable to the 3rd-order SH counterpart, while using fewer SH parameters and enabling faster rendering. Without the pretraining stage, i.e., performing neither pretraining nor distillation, leads to a PSNR drop from 27.12 dB to 26.41 dB, further highlighting the critical role of the pretraining and distillation scheme. We acknowledge that the increased training cost is a limitation of our approach. In future work, we aim to explore strategies to accelerate training while preserving rendering quality and real-time inference performance on mobile devices.

---

### Official Review · Reviewer_sRjw · 2025-10-30

**Soundness:** 3
**Presentation:** 3
**Contribution:** 3
**Rating:** 6
**Confidence:** 4

**Summary:**

Mobile-GS introduces the first real-time 3D Gaussian Splatting framework optimized for mobile GPUs by eliminating costly depth-sorting through a depth-aware, order-independent rendering scheme.
Compared with the prior sorting-free Gaussian representation (SortFreeGS), Mobile-GS models view-dependent opacity and rendering weights in an implicit and lightweight manner, reducing the per-Gaussian parameter footprint. It further integrates spherical-harmonics distillation, neural vector quantization, and contribution-based pruning to enhance compactness and efficiency.
The method achieves ~125 FPS on a Snapdragon 8 Gen 3 device with only 4–5 MB of storage, while preserving visual fidelity comparable to full 3DGS.

**Strengths:**

1. Real-time mobile performance: Demonstrates the first 3D Gaussian Splatting system achieving real-time rendering on mobile GPUs such as Snapdragon 8 Gen 3.
2. Order-independent efficiency: The proposed depth-aware order-independent rendering removes the costly sorting step, significantly improving runtime without significant quality loss.
3. Implicit view modeling: Replaces explicit per-Gaussian weights and opacity with a shared lightweight MLP, enabling stable training and reduced parameters compared with SortFreeGS.
4. Compact design: Through spherical-harmonics distillation, neural vector quantization, and contribution-based pruning, the model compresses storage from hundreds of MB to only ~4–5 MB while maintaining comparable visual fidelity to 3DGS.

**Weaknesses:**

1. Extended training time: The use of a pre-trained teacher model for spherical-harmonics distillation doubles the overall training iterations, increasing computational cost.

2. Complex weighting formulation: The depth-aware weighting term (Eq. 3) appears empirically designed and lacks clear theoretical justification or ablation on its components.

3. Missing key baseline: The paper does not include a direct quantitative comparison with SortFreeGS in Table 2, which should serve as the most relevant baseline for this work.

4. Incomplete dataset coverage (minor): Several Mip-NeRF 360 scenes are missing from the evaluation; including them would strengthen the completeness and reliability of the results.

**Questions:**

1. Pruning mechanism:
 This paper evaluates the contribution of each Gaussian primitive using scale and opacity, while prior works often rely on gradient-based importance or learnable pruning masks. Could the authors discuss the advantages and disadvantages of their design compared to these alternatives, particularly in terms of stability, computational efficiency, and adaptivity during training?
2. Choice of teacher model:
 The framework employs a pre-trained teacher model for spherical-harmonics distillation. Could the authors clarify the motivation behind this choice and whether alternative teacher configurations (e.g., vanilla 3DGS) would impact performance or training cost?
3. Potential use of the teacher model for initialization:
 Since the framework already depends on a teacher model, could this model also be leveraged to provide better Gaussian initialization or coverage, potentially reducing training time?
4. Initialization source:
 Are the initial Gaussian primitives derived from COLMAP point clouds or the teacher model’s reconstructed points?

---

> ### Author Response · Authors · 2025-11-21
> **Response to Reviewer sRjw (Part 1)**
>
> We are grateful for your acknowledgment of the technical novelty and results presented in our work. Please find our response to your questions as follows.
>
> **Q1: The training time and motivation of distillation.**
>
> A1: We acknowledge that incorporating pretraining and distillation introduces additional training time, which is indeed one of the limitations of our proposed Mobile-GS. Nonetheless, our primary goal is to design a real-time rendering method that is practical for mobile deployment while maintaining rendering quality comparable to the original 3DGS. Achieving real-time efficiency and high fidelity necessitates the use of a pretraining-and-distillation pipeline, which effectively enhances rendering performance and reduces the overall storage footprint.
>
> | Method | Mobile-GS (1st-order SH distillation) | 2nd-order SH distillation | 3rd-order SH | 1st-order SH w/o pretraining |
> | --- | --- | --- | --- | --- |
> | PSNR $\uparrow$ | 27.12 | 27.13 | 27.15 | 26.16 |
> | FPS $\uparrow$ | 1125 | 917 | 841 | 1125 |
> | Storage $\downarrow$ | 4.6 MB | 7.3 MB | 9.6 MB | 4.6 MB |
>
> As shown in the Table above, we report results on the Mip-NeRF 360 dataset. Mobile-GS equipped with first-order SH after distillation achieves PSNR that is nearly equivalent to the third-order SH counterpart, while simultaneously providing faster rendering speed with a noticeably smaller parameter set. In contrast, directly training a first-order SH Gaussian Splatting model *without* pretraining and distillation results in an obvious performance drop from 27.12 dB to 26.16 dB, highlighting the insufficiency of naïve low-order SH training.
>
> These findings underscore that the proposed pretraining and distillation pipeline is not merely beneficial but crucial for preserving high-quality novel-view synthesis while enabling effective model compression suitable for mobile platforms. In our future work, we plan to develop more efficient training paradigms and optimization strategies that further reduce training time without compromising the real-time rendering capability and lightweight design of Mobile-GS.
>
> **Q2: The ablation study and motivation on Eq.3 components.**
>
> A2: Thank you for your insightful suggestion. We provide the ablation results for removing $d_i$ and $s_{max}$ on the Mip-NeRF 360 dataset in the Table below.
>
> | Method | Mobile-GS | w/o $d_i$ in Eq.3  | w/o $s_{max}$ in Eq.3  |
> | --- | --- | --- | --- |
> | PSNR $\uparrow$ | 27.12 | 27.03 | 27.08 |
>
> The results clearly indicate that both $d_i$ and $s_{max}$ play critical roles in our rendering mechanism and are essential for achieving high-quality rendering. In particular, eliminating either term leads to a noticeable degradation in performance. This is because depth and scale serve as explicit and interpretable factors in the Gaussian weighting calculation, particularly under our order-independent rendering formulation. Intuitively, Gaussians that lie farther from the camera usually contribute less, while those with larger scale often capture more important structural or appearance information. By incorporating both depth and scale attributes into our rendering process, Mobile-GS can more accurately estimate the contribution of each Gaussian, thereby producing more stable and faithful results even in complex, geometry-overlapping regions. These findings further validate the necessity of these attributes in our contribution and justify our overall rendering design.
>
> **Q3: Missing SortFreeGS in Table 2.**
>
> A3: Thank you for your comment. We have added SortFreeGS in Table 2 of the revised version. We show these results on the Mip-NeRF 360 dataset in the Table below. Since the original SortFreeGS does not involve a quantization process, it is not suitable to be deployed on resource-restricted mobile platforms. SortFreeGS* means a quantified version by our implementation. Compared with existing state-of-the-art methods, our Mobile-GS achieves better rendering quality and real-time inference speed with fewer storage costs.
>
> | Method | PSNR $\uparrow$ | FPS $\uparrow$ | Storage $\downarrow$ | Peak Memory $\downarrow$ | Training $\downarrow$ |
> | --- | --- | --- | --- | --- | --- |
> | 3DGS* | 27.01 | 8 | 61.8 MB | 9489 MB | 0.5 h |
> | Mini-Splatting* | 27.02 | 12 | 36.9 MB | 5841 MB | 0.4 h |
> | Speedy-Splat | 26.92 | 19 | 79.5 MB | 6647 MB | 0.4 h |
> | HAC | 26.98 | 12 | 11.8 MB | 7591 MB | 0.7 h |
> | LocoGS-S | 27.02 | 17 | 8.5 MB | 7018 MB | 0.8 h |
> | C3DGS | 27.03 | 14 | 30.6 MB | 7423 MB | 0.6 h |
> | GES | 26.98 | 18 | 29.4 MB | 7146 MB | 0.7 h |
> | SortFreeGS* | 26.74 | 24 | 64.3 MB | 9628 MB | 1.3 h |
> | Mobile-GS (**Ours**) | 27.12 | 127 | 4.6 MB | 4865 MB | 1.5 h |

---

> > ### Author Response · Authors · 2025-11-21
> > **Response to Reviewer sRjw (Part 2)**
> >
> > **Q4: Incomplete dataset coverage for Mip-NeRF 360**
> >
> > A4: Thank you for your thoughtful advice. We have covered all scenes on the Mip-NeRF 360 dataset and updated the results in Table 1. Our proposed Mobile-GS can achieve 27.12 dB PSNR, surpassing previous lightweight GS methods with fewer storage costs and faster rendering speed.
> >
> > **Q5: Discussion of importance-based and mask-based pruning methods.**
> >
> > A5: Thank you for your insightful feedback. Unlike previous pruning approaches, our proposed contribution-based pruning strategy can be seamlessly integrated into existing pruning methods as a *second-stage* pruning mechanism. This enables finer-grained elimination of redundant Gaussians beyond what coarse pruning methods can achieve. Specifically, our Mobile-GS framework first adopts Mini-Splatting (importance sampling) to perform an initial coarse pruning of redundant Gaussians. We then initialize Mobile-GS using the pretrained Mini-Splatting model. After this coarse reduction, our contribution-based pruning is applied to further refine the Gaussians by accurately identifying and removing low-contribution Gaussians while maintaining rendering fidelity.
> >
> > MaskGaussian represents another category of pruning approaches based on learnable pruning masks. It introduces a binary mask within the traditional volume rendering, where each Gaussian is assigned a learnable mask. We compare MaskGaussian, Mini-Splatting, and the variants combined with our pruning strategy on the Mip-NeRF 360 dataset, and the results are summarized in the Table below.
> >
> > Each pruning method exhibits distinct advantages and limitations:
> >
> > - **MaskGaussian**
> >     - *Advantage:* Learns a binary mask for each Gaussian, offering flexible and data-driven pruning.
> >     - *Limitations:*
> >         1. It cannot be directly applied to the different 3DGS variants because it modifies the volume rendering equation.
> >         2. It introduces an additional pruning loss, $ L_m = (\frac{1}{N}\sum_i^NM_i)^2$, which encourages fewer Gaussians but may lead to training instability.
> > - **Mini-Splatting**
> >     - *Advantage:* Maintains training stability by using importance sampling–based pruning. During inference, its rendering process remains fully compatible with standard 3DGS.
> >     - *Limitation:* A Large number of Gaussians still remain.
> > - **Our Contribution-Based Pruning**
> >     - *Advantages:*
> >         1. Can be integrated with a wide range of GS pruning methods to achieve additional Gaussian reduction.
> >         2. Eliminates Gaussians more accurately by leveraging explicit contribution cues.
> >     - *Limitation:* May introduce a slight loss in precision, but this effect is minimal and acceptable.
> >
> > The experimental results in the Table below demonstrate that our contribution-based pruning can be effectively combined with both MaskGaussian and Mini-Splatting, achieving further Gaussian reduction while preserving rendering quality. This confirms that our method provides a flexible and practical enhancement to existing pruning pipelines.
> >
> > | Method | MaskGaussian [1] | MaskGaussian + Our pruning | Mini-Splatting | Mini-Splatting + Our pruning |
> > | --- | --- | --- | --- | --- |
> > | PSNR $\uparrow$ | 27.24 | 27.16 | 27.41 | 27.38 |
> > | Gaussian Number $\times 10^6$ $\downarrow$ | 1.21 | 0.84 | 0.58 | 0.47 |
> >
> > [1] Liu Y, Zhong Z, Zhan Y, et al. Maskgaussian: Adaptive 3d gaussian representation from probabilistic masks[C]//Proceedings of the Computer Vision and Pattern Recognition Conference. 2025: 681-690.

---

> > > ### Author Response · Authors · 2025-11-21
> > > **Response to Reviewer sRjw (Part 3)**
> > >
> > > **Q6: The choice of teacher model (3DGS)**
> > >
> > > A6: Thank you for your valuable comment. We select Mini-Splatting as the teacher model for our distillation framework because it incorporates a built-in Gaussian pruning mechanism that effectively removes low-importance Gaussians, thereby achieving substantially lower storage costs while preserving high rendering quality. To validate this choice, we conduct experiments on the Mip-NeRF 360 dataset using either the original 3DGS or Mini-Splatting as the teacher model, as shown in the Table below.
> > >
> > > | Method | 3DGS | 3DGS + Our 1st-order SH disitllation | Mini-Splatting | Mini-Splatting + Our 1st-order SH disitllation |
> > > | --- | --- | --- | --- | --- |
> > > | PSNR $\uparrow$ | 27.24 | 27.22 | 27.38 | 27.37 |
> > > | Storage $\downarrow$ | 839.9 MB | 371.6 MB | 137.4 MB | 94.5 MB |
> > >
> > > Mini-Splatting reconstructs 3D scenes with significantly fewer Gaussian parameters and reduced storage requirements, while maintaining competitive rendering accuracy. By integrating our proposed first-order SH distillation into these teacher models, we further compress the representation by reducing redundant SH parameters, which leads to an even more compact Gaussian model without compromising fidelity.
> > >
> > > Notably, with our distillation strategy, the student models using 1st-order SH achieve PSNR values that are highly comparable to their 3rd-order SH teachers, regardless of whether the teacher is 3DGS or Mini-Splatting. This highlights the effectiveness of the proposed distillation mechanism, which enables the student model to retain the essential radiance information despite having substantially fewer SH coefficients.
> > >
> > > Overall, these results prove that Mini-Splatting serves as an efficient and practical teacher model for Mobile-GS, and demonstrate that our first-order SH distillation significantly enhances model compactness while preserving high-quality rendering performance.
> > >
> > > **Q7: Potential use of the teacher model for initialization to reduce training time.**
> > >
> > > A7: Thank you for your insightful suggestion. Actually, in our pipeline, the teacher model is used to initialize the student model. This initialization facilitates a stable and effective distillation process, where the objective is to transfer the representation capacity of the original 3rd-order SH representation (with N×48 parameters) into a significantly more compact 1st-order SH representation (with N×12 parameters). The goal is to preserve rendering fidelity that is comparable to the 3rd-order SH model while achieving substantial parameter reduction.
> > >
> > > At present, our design primarily focuses on ensuring high-quality distillation and does not explicitly target training time optimization. We acknowledge that incorporating mechanisms to reduce training time would further improve the practicality of Mobile-GS, especially for large-scale or real-time applications. In future work, we plan to explore more efficient training paradigms and distillation strategies that can accelerate convergence while maintaining or potentially improving the current rendering performance.
> > >
> > > **Q8: Are the initial Gaussian primitives derived from COLMAP point clouds or the teacher model’s reconstructed points?**
> > >
> > > A8: Our training process can be divided into pre-training and fine-tuning. In the pre-training stage, we train the teacher GS with COLMAP point clouds as initialization. In the fine-tuning stage, we initialize the student GS with the parameters of the teacher GS, such as Gaussian position, rotation, scale, and First-order SH parameters.

---

### Official Review · Reviewer_Jsav · 2025-10-31

**Soundness:** 4
**Presentation:** 3
**Contribution:** 4
**Rating:** 6
**Confidence:** 4

**Summary:**

This paper proposes a mobile-friendly 3D Gaussian Splatting pipeline that removes depth sorting via depth-aware order-independent rendering, then recovers quality with a lightweight, view-dependent enhancement MLP. It further compresses the representation using first-degree SH distillation, neural vector quantization, and contribution-based pruning to cut storage to a few MB while keeping fidelity. Experiments report >100 FPS on Snapdragon 8 Gen 3 and >1k FPS on RTX 3090 with competitive quality versus 3DGS and recent lightweight baselines.

**Strengths:**

- This paper claims per-tile sorting as the dominant bottleneck and introduces a simple, parallelizable order-independent blending scheme to remove it.
- A small view-conditioned MLP effectively suppresses transparency/occlusion artifacts that arise from sorting-free compositing.
- The compression stack (first-degree SH distillation + neural vector quantization + contribution-based pruning) is complementary and yields strong storage reductions with limited quality loss.
- The evaluation is extensive, includes ablations/runtime breakdowns, and demonstrates impressive reported throughput on Snapdragon 8 Gen 3.

**Weaknesses:**

- The novelty relative to contemporary sorting-free methods (e.g., SortFreeGS, stochastic/OIT-style splatting) is incremental and would benefit from a deeper theoretical or empirical comparison.
-  Some hyperparameters, such as pruning thresholds/schedules, codebook sizes and SH-order trade-offs need further analysis.
- The related works on network design and pruning should be added.

**Questions:**

See weaknesses.

---

> ### Author Response · Authors · 2025-11-21
> **Response to Reviewer Jsav (Part 1)**
>
> We are grateful for your recognition of our paper results and the effectiveness of our approach. Please find our point-to-point response to the review comments below.
>
> **Q1: Comparisons with other sorting-free methods.**
>
> A1:  Thank you for your insightful suggestion. Compared to other sorting-free approaches, our Mobile-GS provides a more comprehensive and deployment-oriented solution that enables high-quality and real-time rendering on mobile devices. We present detailed comparisons on the Mip-NeRF 360 dataset in the Table below.
>
> | Method | Rendering | Weighting | PSNR$\uparrow$ | Storage$\downarrow$ | FPS$\uparrow$ | How to solve order-ambigous problem |
> | --- | --- | --- | --- | --- | --- | --- |
> | SortFreeGS* | $ \mathbf{C} = \frac{c_{bg} w_{bg} + \sum_{i=1}^{\mathcal{N}} c_i \alpha_i w(d_i) }{ w_{bg} + \sum_{i=1}^{\mathcal{N}} \alpha_i w(d_i)}$ | $w(d_i) = \exp\left(-\sigma d_i^\beta\right)$ | 26.74 | 64.3 MB | 18 | Introducing addtional Sheprical Harmonics (SH) parameters |
> | GES | $ \mathbf{C} = \frac{C_sW_s+C_G}{W_s+W_G}$ | $ W_G(\hat{\mathbf{x}})=\sum_{i=1}^N[1(d_i<d_s(\hat{\mathbf{x}})+\epsilon)]\alpha_i(\hat{\mathbf{x}})$ | 27.02 | 29.4 MB | 24 | Using two-stage rendering(1. volume rendering for depth filtering farther Gaussians, 2. rendering remained Gaussians with order-independet rendering) |
> | Ours | $\mathbf{C}=(1-T)\frac{\sum_{i=1}^{\mathcal{N}} c_i \alpha_i w_i} {\sum_{i=1}^{\mathcal{N}} \alpha_{i}w_i  } +T\mathbf{c}_{bg}$ | $ w_i= \phi^2_i + \frac{\phi_i }{d_i^2} + exp(\frac{s_{max}}{d_i})$ | 27.12 | 4.6 MB | 127 | Introducing a lightweight MLP conditioned on Gaussian-camera vetors to enhance 3D spatial understanding ability |
>
> It is worth noting that SortFreeGS* refers to the quantized version of SortFreeGS, as the original method does not include a quantization stage. In terms of PSNR, storage cost, and rendering FPS, our Mobile-GS consistently achieves superior performance over prior sorting-free techniques, such as SortFreeGS [1] and GES [2]. This improvement stems from our integrated design that incorporates quantization, pruning, and a view-dependent enhancement mechanism. With respect to rendering formulations, GES follows a formulation similar to SortFreeGS, whereas our method adopts a transmittance proxy enriched with view-dependent modulation to more effectively capture the underlying 3D scene structure.
>
> The Gaussian weight computation also differs substantially across these approaches. SortFreeGS leverages the Gaussian depth to modulate its contribution but does not account for the Gaussian scale, which we find to be critical. GES, on the other hand, relies on a two-stage rendering. It first renders a depth image using conventional volume rendering and then filters out distant Gaussians by comparing their depths against the rendered depth map for later sorting-free rendering. This two-stage rendering pipeline relies on precise depth rendering and increases computational load, so it is not well-suited for mobile deployment. In contrast, Mobile-GS exploits both depth and scale attributes of each Gaussian to compute an importance weight, reflecting the intuition that farther Gaussians should have lower contribution, while larger Gaussians typically contribute more meaningful rendering information.
>
> Theoretically, A key challenge for sorting-free methods is the potential order ambiguity in regions where geometry overlaps. SortFreeGS attempts to address this by introducing additional spherical harmonics parameters to model view-dependent opacity. However, this design incurs significant overhead and is unfavorable for practical mobile usage. Our Mobile-GS resolves this limitation by enhancing the view-dependent effect through a learnable parameter $\phi$, predicted by a lightweight MLP conditioned on Gaussian attributes. This formulation achieves high-quality rendering without introducing a prohibitive computational or memory burden.
>
> Overall, Mobile-GS is carefully tailored to minimize resource consumption, reduce Gaussian parameter storage, and maintain real-time rendering performance on mobile hardware. We have incorporated this expanded discussion into Appendix D.1 to provide clearer context and justification for our design choices.
>
> [1] Hou Q, Rauwendaal R, Li Z, et al. Sort-free gaussian splatting via weighted sum rendering[J]. arXiv preprint arXiv:2410.18931, 2024.
>
> [2] Ye K, Shao T, Zhou K. When gaussian meets surfel: Ultra-fast high-fidelity radiance field rendering[J]. ACM Transactions on Graphics (TOG), 2025, 44(4): 1-15.

---

> > ### Author Response · Authors · 2025-11-21
> > **Response to Reviewer Jsav (Part 2)**
> >
> > **Q2:  Hyperparameters about pruning thresholds.**
> >
> > A2: In our proposed pruning strategy, we eliminate low-contribution Gaussians based on a predefined threshold. A larger pruning threshold corresponds to more aggressive pruning, resulting in a smaller set of retained Gaussians. To analyze the effect of this threshold on performance, we conduct an ablation study on the Mip-NeRF 360 dataset summarized in the Table below. Specifically, we take Mobile-GS without pruning as the baseline and apply our contribution-based pruning mechanism with thresholds ranging from 0.1 to 0.6.
> >
> > | Threshold | Baseline | 0.1 | 0.2 | 0.4 | 0.6 |
> > | --- | --- | --- | --- | --- | --- |
> > | Num. $\times 10^6$ $\downarrow$ | 0.56 | 0.55 | 0.47 | 0.34 | 0.18 |
> > | PSNR $\uparrow$ | 27.22 | 27.15 | 27.12 | 26.47 | 25.85 |
> > | FPS* $\uparrow$ | 109 | 111 | 127 | 141 | 164 |
> >
> > The results indicate that a threshold of 0.2 provides the most favorable balance between rendering quality and computational efficiency. Lower thresholds retain more Gaussians but offer limited improvement in visual fidelity, whereas higher thresholds remove too many Gaussians and introduce noticeable quality degradation. The threshold of 0.2 achieves an optimal trade-off by effectively reducing redundancy while preserving high-quality rendering performance. Based on these observations, we adopt 0.2 as the default pruning threshold in Mobile-GS. The experimental results and corresponding analysis have been added to Table 5 in the revised manuscript for completeness.
> >
> > **Q3:  Hyperparameters about pruning schedules**
> >
> > A3: In our contribution-based pruning strategy, we remove Gaussians with low estimated contribution according to their scale and opacity attributes. Intuitively, Gaussians with larger opacity and larger scale are expected to contribute more to the final rendered image. To validate the effectiveness of this pruning scheme, we conduct an ablation study on the Mip-NeRF 360 dataset using Mobile-GS without the proposed contribution-based pruning as the baseline. We then evaluate three variants: pruning based solely on opacity, solely on scale, and on the combination of both attributes. The results are summarized in the Table below.
> >
> > | Method | Baseline | Opacity | Scale | Opacity & Scale |
> > | --- | --- | --- | --- | --- |
> > | Num. $\times 10^6$ $\downarrow$ | 0.56 | 0.43 | 0.45 | 0.47 |
> > | PSNR $\uparrow$ | 27.22 | 26.84     | 26.87   | 27.12 |
> > | FPS* $\uparrow$ | 109 | 135  | 132 | 127 |
> >
> > These results show that pruning using only opacity or only scale leads to overly aggressive pruning and substantial degradation in rendering quality for overall performance. In contrast, pruning based on both opacity and scale yields a significantly better balance. It reduces the number of Gaussians at a moderate rate while maintaining high rendering fidelity. This improvement arises because the joint consideration of opacity and scale provides a more reliable estimate of each Gaussian’s actual contribution, thereby preventing the elimination of Gaussians that are structurally or visually important.
> >
> > Overall, these findings confirm that incorporating both attributes into our pruning schedule enables more accurate identification of low-contribution Gaussians and produces a more effective and robust pruning strategy. The corresponding results and analysis have been included in the revised manuscript.
> >
> > **Q4: Hyperparameters about codebook sizes**
> >
> > A4: Thank you for your insightful suggestion. Our proposed neural vector quantization employs K-means clustering to encode Gaussian parameters into a compact codebook. The size of this codebook directly influences both rendering performance and storage efficiency. A larger codebook provides more clusters, thereby reducing quantization error but increasing storage overhead. Conversely, a smaller codebook reduces storage costs but may introduce noticeable precision loss. To systematically examine this trade-off, we conduct experiments on the Mip-NeRF 360 dataset with different codebook sizes, ranging from $2^6$ to $2^{12}$, as reported in the Table below.
> >
> > | Codebook size | $2^6$ | $2^8$ | $2^{10}$ | $2^{12}$ |
> > | --- | --- | --- | --- | --- |
> > | PSNR $\uparrow$ | 25.52 | 26.83 | 27.12 | 27.15 |
> > | Storage  $\downarrow$ | 3.84 MB | 4.2 MB | 4.6 MB | 7.9 MB |
> >
> > The results show that very small codebooks, such as $2^6$, lead to substantial degradation in PSNR due to excessive quantization. On the other hand, very large codebooks, such as $2^{12}$, require significantly more storage, which contradicts our goal of designing a lightweight model for mobile deployment. Based on this analysis, we select a codebook size of $2^{10}$, which achieves an effective balance between compact storage and high-fidelity rendering. This configuration minimizes precision loss while avoiding unnecessary storage consumption. The corresponding experimental results and justification have been added to the revised manuscript for clarity.

---

> > > ### Author Response · Authors · 2025-11-21
> > > **Response to Reviewer Jsav (Part 3)**
> > >
> > > **Q5: Ablation study about SH-order trade-offs**
> > >
> > > A5: Thank you for your valuable advice. Empirically, using higher-order SH parameters enhances the representation capacity of the model and generally leads to improved rendering accuracy. However, this comes at the cost of significantly increased parameters and storage overhead, which is undesirable for mobile deployment. To address this trade-off, we introduce Mobile-GS equipped with a first-order SH distillation strategy. The goal is to substantially reduce the number of SH parameters while mitigating the corresponding performance drop. We conduct an ablation study on different SH orders on the Mip-NeRF 360 dataset, and the results are summarized in the Table below.
> > >
> > > | Method | Mobile-GS (1st-order SH distillation) | 0th-order SH distillation | 2nd-order SH distillation | 3rd-order SH  |
> > > | --- | --- | --- | --- | --- |
> > > | PSNR $\uparrow$ | 27.12 | 27.04 | 27.13 | 27.15 |
> > > | FPS $\uparrow$ | 1125 | 1219 | 917 | 841 |
> > > | Storage $\downarrow$ | 4.6 MB | 3.6 MB | 7.3 MB | 9.6 MB |
> > >
> > > With the proposed distillation scheme, Mobile-GS using first-order SH achieves rendering quality that is remarkably close to that of the third-order SH model. This improvement is primarily attributed to our carefully designed distillation loss, which provides accurate supervisory guidance from a powerful teacher model. These findings confirm the effectiveness of our SH distillation approach. It preserves high rendering fidelity while greatly reducing model size. In contrast, further reducing the SH order to zero-order introduces substantial degradation in image quality, indicating that 0th-order SH lacks sufficient representational power even with distillation. Therefore, we adopt first-order SH in Mobile-GS as the optimal choice, achieving a strong balance between rendering quality, model compactness, and inference speed.
> > >
> > > **Q6: The related works on network design and pruning should be added.**
> > >
> > > A6: Thank you for your constructive suggestion. In our revised version, we have added more references [3-10] about network design and pruning in the section on Related Work. These references are shown below. If we neglect some references, please feel free to let us know.
> > >
> > > [3] Höllein L, Božič A, Zollhöfer M, et al. 3dgs-lm: Faster gaussian-splatting optimization with levenberg-marquardt[C]//Proceedings of the IEEE/CVF International Conference on Computer Vision. 2025: 26740-26750.
> > >
> > > [4] Kheradmand S, Rebain D, Sharma G, et al. 3d gaussian splatting as markov chain monte carlo[J]. Advances in Neural Information Processing Systems, 2024, 37: 80965-80986.
> > >
> > > [5] Kulhanek J, Rakotosaona M J, Manhardt F, et al. LODGE: Level-of-Detail Large-Scale Gaussian Splatting with Efficient Rendering[J]. arXiv preprint arXiv:2505.23158, 2025.
> > >
> > > [6] Liu Y, Zhong Z, Zhan Y, et al. Maskgaussian: Adaptive 3d gaussian representation from probabilistic masks[C]//Proceedings of the Computer Vision and Pattern Recognition Conference. 2025: 681-690.
> > >
> > > [7] Zhang Y, Jia W, Niu W, et al. GaussianSpa: An" Optimizing-Sparsifying" Simplification Framework for Compact and High-Quality 3D Gaussian Splatting[C]//Proceedings of the Computer Vision and Pattern Recognition Conference. 2025: 26673-26682.
> > >
> > > [8] Tang Z, Feng C, Cheng X, et al. NeuralGS: Bridging Neural Fields and 3D Gaussian Splatting for Compact 3D Representations[J]. arXiv preprint arXiv:2503.23162, 2025.
> > >
> > > [9] Ren K, Jiang L, Lu T, et al. Octree-gs: Towards consistent real-time rendering with lod-structured 3d gaussians[J]. arXiv preprint arXiv:2403.17898, 2024.
> > >
> > > [10] Li H, Liu J, Sznaier M, et al. 3D-HGS: 3D Half-Gaussian Splatting[C]//Proceedings of the Computer Vision and Pattern Recognition Conference. 2025: 10996-11005.

---

### Official Review · Reviewer_JUSS · 2025-11-02

**Soundness:** 3
**Presentation:** 3
**Contribution:** 3
**Rating:** 6
**Confidence:** 3

**Summary:**

This paper proposes Mobile‑GS, a 3D Gaussian Splatting pipeline designed for real‑time rendering on mobile devices. The key contributions include:

1) Depth‑aware order‑independent rendering (OIR) that removes near‑to‑far sorting by blending all Gaussians affecting a pixel with a depth/scale‑modulated weight; the weight includes an MLP‑predicted, view‑dependent factor ϕ. A small neural view‑dependent opacity/weighting module combats transparency artifacts that arise from dropping strict alpha compositing.

2) Compression for mobile: (i) first‑degree SH distillation from a teacher (Mini‑Splatting) to reduce color parameters, (ii) neural vector quantization using sub‑codebooks plus tiny decoders for diffuse/view‑dependent SH components, and (iii) contribution‑based pruning guided by opacity and maximum scale.

3) Implementation & results: a Vulkan implementation on a Snapdragon 8 Gen 3 device; the method achieves 116–127 FPS at mobile resolutions with ~4–5 MB per‑scene storage, and >1,100 FPS on an RTX 3090 with similar or better quality than lightweight baselines. The paper identifies sorting as the desktop bottleneck and provides runtime breakdowns showing the MLP overhead is modest.

**Strengths:**

++ The paper starts from a concrete performance study showing that near‑to‑far sorting dominates 3DGS inference time. It then replaces sorting with a depth‑aware, order‑independent blend: per‑pixel colors are computed by normalizing a weighted sum over all contributing Gaussians, where the weights increase with proximity and scale and are modulated by a small learned, view‑dependent factor.

++ Order‑independent blending can cause depth‑ambiguity/“see‑through” artifacts; the authors respond with a tiny opacity/weighting MLP that conditions on Gaussian geometry, SH appearance and view direction to predict 𝜙 and a view‑conditioned opacity. The ablation in Table 3 shows that removing this module causes a notable quality drop (e.g., PSNR from 28.45 → 28.06 on Mip‑NeRF360) while the runtime overhead is small.

++ Three components—first‑degree SH distillation, neural vector quantization (NVQ) with sub‑codebooks and tiny decoders, and contribution‑based pruning—work together to shrink the footprint while keeping quality.

**Weaknesses:**

-- Eq. (2) uses a global transmittance 𝑇, then defines 𝑇, which is order‑dependent and index‑ambiguous under OIR; this needs a precise approximation/implementation

-- Fig. 3 claims tile‑based rasterization is removed and “all Gaussians associated with a pixel” are blended, but the paper doesn’t detail how per‑pixel lists are built/cached on GPU (desktop or mobile).

-- The proposed weight 𝑤_i (Eq. (3), pp. 4–5) blends squared, inverse‑squared and exponential terms whose dynamic ranges can differ by orders of magnitude. The paper shows an ablation for turning OIR on/off (Table 3), but not a component‑level analysis, nor a discussion of clipping/normalization. Without this, it’s hard to assess numerical stability, generalization to thin structures, and the sensitivity to scene scale.

**Questions:**

1. Please clarify the exact computation of T in Eq. (2). If it is not the product in the definition (which requires sorting), what approximation is actually used and how is it implemented? Is it related to weighted blended OIT (e.g., a transmittance proxy from aggregated α)? A small derivation or pseudocode would help.

2. What data structure replaces tile binning? Are you using screen‑space bounding ellipses with per‑pixel lists, hierarchical culling, or compute‑shader binning? Please quantify the cost of building these lists, especially on mobile.

3. What ranges are enforced for 𝑑_𝑖 and 𝑠_max? Are terms clamped or normalized per‑tile/view? Could you share an ablation removing each term?

4. What exact resolution(s) and camera path were used for Table 2? How long were runs and what was the steady‑state FPS after 5–10 minutes? Any power draw measurements?

---

> ### Author Response · Authors · 2025-11-21
> **Response to Reviewer JUSS (Part 1)**
>
> Thank you for the kind words for recognizing the impact of our proposed Mobile-GS. Please find our point-to-point response to your questions below.
>
> **Q1: How to calculate the global transmittance 𝑇. Is it order‑dependent and index‑ambiguous under OIR?**
>
> A1:  Thank you for your valuable comment. Our transmittance is neither order-dependent nor index-ambiguous under OIR due to the use of our proposed view-dependent opacity.
>
> Specifically, our global transmittance *T* is computed through $ T = \prod_{j=1}^{N} (1 - \alpha_j)$ as mentioned below Eq. 2. Our transmittance *T*  inherently contains view-dependent information. This is because *T* is obtained from $\alpha$ and $\alpha$ is derived from the opacity via $\alpha_i = o_i \exp\left(-\frac{1}{2}\Delta x_i^T\Sigma_i^{-1}\Delta x_i\right)$. Our view-dependent opacity is predicted by a lightweight MLP conditioned on viewing direction and other Gaussian attributes, as shown in Figure 4. With this opacity, our transmittance is not index-ambiguous, since it contains view-dependent information from the view-dependent opacity. Without this view-dependent opacity, it would cause the mentioned index-ambiguous problem, and the rendering performance would degrade significantly, as shown in Figure 7 and the ablation study of Table 3. Since we do not have a sorting process within the rendering process,  we exploit the opacity with the view-dependent information to mitigate this problem. This opacity enhances the transimittance to understand the foreground, background, and the 3D spatial structure among objects.
>
> Consequently, our **view-dependent enhancement strategy** is crucial for achieving order-independent rendering, enabling our approach to approximate the effects of traditional depth-sorting techniques while maintaining high efficiency and fidelity. This design not only mitigates the limitations associated with order-independent rendering but also allows for accurate and visually consistent scene reconstruction across varying viewpoints.
>
> **Q2: How to replace the original tile-based rendering and data structure**
>
> A2: In our CUDA implementation, the rendering pipeline is organized into two primary stages: **RenderNoSortKernel** and **ComposeKernel**. The **RenderNoSortKernel** is responsible for computing the color contribution of each Gaussian primitive in parallel for its corresponding pixels. Specifically, for each Gaussian, we first project the 3D Gaussian primitive into the 2D pixel space and determine its axis-aligned bounding box to identify the set of pixels that it influences. Subsequently, we compute the contribution of the Gaussian to each of these pixels and perform per-pixel accumulation of the resulting color values.
>
> Following this, the **ComposeKernel** stage is executed, which combines the accumulated foreground colors with the background colors on a per-pixel basis, also in parallel. This stage ensures that the final rendered image accurately reflects both foreground and background components according to the computed transmittance and opacity values. The overall workflow is illustrated in the CUDA pseudocode below, which clarifies the step-by-step execution of our rendering process. We build the pixel list through
>
> ```jsx
>  torch::Tensor out_color = torch::full({NUM_CHANNELS, H, W}, 0.0, float_opts);
> ```
>
> The cost of building this list is about `C * H * W * sizeof(float)` bytes, which depends on the resolution of rendering images. We have added more details about this process in our revised version to make it clearer.
>
> ```python
> ## RenderNoSortKernel
>
> for each Gaussian g in parallel:
>
>     float weight = phi*phi + phi / (depth * depth) + expf(max_scale / depth);
>
>     compute pixel bounding box ## the coverage of Gaussian projected on pixel
>
>     for each pixel p in bounding box:
>
>         float dx = (center.x - px);
>         float dy = (center.y - py);
>
>         float power = -0.5f * (co.x * dx * dx + co.z * dy * dy) - co.y * dx * dy;
>         float alpha = fminf(0.99f, co.w * __expf(power));
>
>         float logTerm = logf(fmaxf(1.0f - alpha, 1e-6f));
>
>         atomicAdd(&T[p], logTerm);
>         atomicAdd(&out_color[p], c * alpha * weight);
>         atomicAdd(&w_fg[p], alpha * weight);
> ```
>
> ```python
> ## ComposeKernel
>
> for each pixel p in parallel:
>     Wfg = w_fg[p] + eps
>     T   = exp(T[p])
>
>     out_color[p] = (out_color[p] / Wfg) * (1 - T) + T * bg_color
> ```

---

> > ### Author Response · Authors · 2025-11-21
> > **Response to Reviewer JUSS (Part 2)**
> >
> > **Q3: A component‑level analysis and discussion of normalization, range, stability**,  **and ablation for Eq.3**
> >
> > A3: In our rendering formulation, we do not impose explicit constraints on the range of the Gaussian depth $d_i$ or the maximum scale $s_{max}$. This design choice is motivated by the fact that both the depth and scale of Gaussians naturally vary according to the camera parameters. Imposing arbitrary limits on these attributes could undesirably constrain the growth and representation of 3D Gaussians, potentially degrading the fidelity of the reconstruction.
> >
> > Furthermore, we do not perform normalization of $ d_i $ or $s_{max}$ during the computation of Gaussian weights in Eq. 3, as such normalization has a negligible impact on rendering performance. From a theoretical perspective, in the calculation of the foreground color, $ \frac{\sum_{i=1}^{\mathcal{N}}c_i \alpha_i w_i}{\sum_{i=1}^{\mathcal{N}}\alpha_{i}w_i} $, we divide by $\sum_i \alpha_i w_i$ to ensure numerical stability and prevent potential overflow during accumulation. This guarantees that the color computation remains stable even when dealing with Gaussians of varying scales and depths.
> >
> > To empirically validate this design choice, we conducted experiments comparing the rendering quality with and without normalization of $s_{max}$ and $d_i$. The results, summarized in the Table below, demonstrate that normalization of these terms does not produce significant improvements in rendering quality, thereby justifying the inessentiality of such normalization in our method.
> >
> > | Method | Mobile-GS | Normalized $s_{max}$  | Normalized $d_i$  |
> > | --- | --- | --- | --- |
> > | PSNR $\uparrow$ | 27.12 | 27.11 | 27.13 |
> >
> > As for the ablation study about removing these terms, we show these results below:
> >
> > | Method | Mobile-GS | w/o $d_i$ in Eq.3  | w/o $s_{max}$ in Eq.3  |
> > | --- | --- | --- | --- |
> > | PSNR $\uparrow$ | 27.12 | 27.03 | 27.08 |
> >
> > It is evident that the Gaussian depth $d_i$ and maximum scale $s_{max}$ play a critical role in our rendering framework. Omitting these attributes results in a noticeable degradation of rendering performance. This is because both depth and scale can act as direct factors for the Gaussian weight contributions. For instance, Gaussians that are farther from the camera inherently contribute less to the final color, while larger Gaussians have a more important influence on the rendered image. By explicitly incorporating these attributes into the rendering process, our method is able to more accurately model the contributions of individual Gaussians, thereby enhancing the fidelity and visual quality of the resulting renderings. Moreover, the inclusion of $d_i$ and $s_{max}$ enables our approach to naturally account for geometric and photometric variations across the scene, ensuring that both near and far, small and large primitives are appropriately weighted. This contributes to stable and high-quality rendering, particularly in complex scenes with varying Gaussian distributions.
> >
> > **Q4: Is transmittance 𝑇 related to weighted blended OIT?**
> >
> > A4: Thank you for your insight. Our transmittance is calculated through  $ T = \prod_{j=1}^{N} (1 - \alpha_j)$, which differs fundamentally from the weighted blended Order-Independent Transparency (OIT) approaches that rely on aggregated $\alpha$ values. While both formulations are order-independent, this is the only aspect they share.
> >
> > To further clarify, we provide a comparison of various transmittance proxy calculations in the Table below. Our approach is designed to closely resemble the transmittance computation in the original 3D Gaussian Splatting (3DGS) framework. Ours offers the significant advantage of being **sorting-free**. Unlike 3DGS, where the transmittance is applied directly to each Gaussian, our formulation operates at the per-pixel level, modulating the foreground and background contributions. With our proposed view-dependent enhancement, our transmittance allows for efficient and accurate rendering without requiring explicit sorting of primitives, while still preserving the fidelity.
> >
> > | Method | Meshkin [1] | Bavoil et al. [2] | Ours |
> > | --- | --- | --- | --- |
> > | Transmittance Proxy | $ 1-\sum_i^N \alpha_i $ | $ [1- \frac{1}{N} \sum_i^N \alpha_i]^N$ |  $ \prod_{j=1}^{N} (1 - \alpha_j)$ |
> >
> > [1] Meshkin H. Sort-independent alpha blending[J]. GDC Talk, 2007, 2(4).
> >
> > [2] Bavoil L, Myers K. Order independent transparency with dual depth peeling[J]. NVIDIA OpenGL SDK, 2008, 1(12): 2-4.
> >
> > **Q5: What exact resolution(s) and camera path were used for Table 2?**
> >
> > A5: For Table 2, we report results in 1600 × 1063 resolution, following the default setting of 3DGS. Except for PSNR calculation, we render novel views in an ellipse camera path around the centered object. The rendering display can be found at the bottom of our project page.

---

> ### Author Response · Authors · 2025-11-21
> **Response to Reviewer JUSS (Part 3)**
>
> **Q6: How long were run and what was the steady‑state FPS after 5–10 minutes?**
>
> A6: The reported FPS in our paper is the cold-start FPS, which means we directly test FPS without warming up. For a more comprehensive evaluation, we heat up the mobile with the Snapdragon 8 Gen 3 GPU for 10 minutes through 3DGS rendering and test the steady‑state FPS on the Mip-NeRF 360 dataset.  The results are shown below. On mobiles, FPS **drops over time** because of thermal throttling, power limits, GPU clock downscaling, and NPU/CPU frequency limits. We can find that our Mobile-GS can still achieve 74 Steady‑state FPS for real-time rendering on mobiles. We have added this evaluation to our revised version.
>
> | Method | 3DGS | Speedy-Splat | SortFreeGS | Mobile-GS (Ours) |
> | --- | --- | --- | --- | --- |
> | Cold-start FPS $\uparrow$ | 8 | 19 | 24 | 127 |
> | Steady‑state  FPS $\uparrow$ | 3 | 10 | 18 | 74 |
>
> **Q7: Power draw measurements?**
>
> A7: Thank you for your insightful suggestion. We leverage Qualcomm Trepn Profiler to evaluate the power (W) of Vulkan operators and MLP on the Mip-NeRF 360 dataset in 1600 x 1063 resolution, comparing our proposed Mobile-GS with 3DGS* and SortFreeGS*. 3DGS* and SortFreeGS* mean their quantized versions for deployment on the mobile with Snapdragon 8 Gen 3 GPU. The power measurement results are shown below. As shown in this Table, our Mobile-GS uses the least power compared to 3DGS and SortFreeGS, which demonstrates the practicality of our method. We have added this comparison in our revised version.
> | Method | Preprocessing | Sorting | MLP | Rasterization | Total |
> | --- | --- | --- | --- | --- | --- |
> | 3DGS* | 1.64 | 2.09 | 0 | 2.16 | 5.89 |
> | SortFreeGS* | 1.78 | 0 | 0 | 2.25 | 4.03 |
> | Mobile-GS | 0.17 | 0 | 0.24 | 0.42 | 0.83 |

---

### Author Response · Authors · 2025-11-21
**General Response**

We would like to express our sincere gratitude to all the reviewers for their thoughtful and constructive feedback on our submission. We genuinely appreciate the time and effort they invested in helping us refine our work.

It is encouraging that all the reviewers recognize and agree that our proposed Mobile-GS achieves real-time and high-quality performance on Mobiles and presents "highly impressive", "highly promising" results through "effective and convincing" techniques.  We believe our research can push forward the development of real-time 3DGS rendering on mobile devices. We also confirm that we will release code upon acceptance to foster more real-time rendering research on mobiles and ensure full reproducibility. Comprehensive documentation and usage instructions will be provided to facilitate adoption and replication by the community.

We sincerely appreciate all constructive suggestions, which have been invaluable in improving the clarity and presentation of the paper. In response, we have revised the manuscript to address and clarify the raised concerns. In summary, the paper has been improved in the following aspects:

- Adding a more detailed description for the global transmittance in Section 3.1
- Adding a more precise description for the rendering process in Figure 3
- Adding component‑level ablation studies for Eq. 3 in Table 3
- Adding power draw and steady-state FPS measurements in Appendix G
- Adding more discussion with other sorting-free methods in Appendix D.1
- Adding more pruning analysis in Section 4.1
- Adding ablation studies for SH-order trade-offs in Table 3
- Adding more references in the Related Work section
- Adding the adaptivity experiment of the proposed pruning in Table 7
- Adding SortFreeGS and training time in Table 2

We believe these improvements further elevate the quality and clarity of our paper, and we look forward to further review feedback.

---

### Meta-Review · Area_Chair_pfjC · 2026-01-06

**Summary:**

The reviewers recognized Mobile-GS as a practical contribution to the deployment of 3D Gaussian Splatting (3DGS) on mobile devices. The paper received initial scores of (6, 6, 6, 6, 4). While reviewers initially questioned the theoretical soundness of the Order-Independent Rendering (OIR) formulation and the fairness of baseline comparisons, the rebuttal provided robust clarifications and new empirical data. The consensus is that the framework effectively bridges the gap between high-fidelity 3D rendering and mobile constraints. Given the effective rebuttal and the addition of real-world mobile metrics, I recommend acceptance of the paper.

**Reviewer Concerns:**

Addressed concerns:
- Theoretical validity: the authors clarified the global transmittance calculation using commutative log-sum accumulation and justified the weighting terms through new ablation studies.
- Baseline comparisons: new quantitative comparisons against SortFreeGS (Quantized) demonstrated Mobile-GS’s clear superiority in both storage (4.6MB vs. 64MB) and speed (127 FPS vs. 24 FPS).
- Mobile metrics: the authors added steady-state FPS and power consumption data (0.83W), providing necessary evidence of the system's viability under thermal constraints.
- Distillation necessity: an ablation study confirmed that training without the distillation pipeline results in a significant PSNR drop (~1 dB), justifying the training complexity.

Concerns might still be outstanding:
- Training efficiency: the extended training time (~1.5 hours) due to the teacher-student distillation process remains an acknowledged limitation, though acceptable for the target inference-heavy use case.

**Reviewer Scores:**

The manuscript received an initial score of (6, 6, 6, 6, 4). Following the rebuttal, the major technical and fairness concerns were resolved. It is expected that the majority of reviewers (JUSS, Jsav, sRjw, xVP3) will maintain their positive stance. Also, as mentioned in the initial review, Reviewer JKAA is likely to increase their score from 4 to 6, leading to a clear consensus for acceptance.

---

### Decision · Program_Chairs · 2026-01-26

Accept (Poster)